# Scaling In-the-Wild Training for Diffusion-based Illumination Harmonization and Editing by Imposing Consistent Light Transport

**Lvmin Zhang[1], Anyi Rao[2], Maneesh Agrawala[1]**
[1]Stanford University, [2]Hong Kong University of Science and Technology
{lvmin,maneesh}@cs.stanford.edu, anyirao@ust.hk

## Abstract

Diffusion-based image generators are becoming unique methods for illumination harmonization and editing. The current bottleneck in scaling up the training of diffusion-based illumination editing models is mainly in the difficulty of preserving the underlying image details and maintaining intrinsic properties, such as albedos, unchanged. Without appropriate constraints, directly training the latest large image models with complex, varied, or in-the-wild data is likely to produce a structure-guided random image generator, rather than achieving the intended goal of precise illumination manipulation. We propose Imposing Consistent Light (IC-Light) transport during training, rooted in the physical principle that the linear blending of an object's appearances under different illumination conditions is consistent with its appearance under mixed illumination. This consistency allows for stable and scalable illumination learning, uniform handling of various data sources, and facilitates a physically grounded model behavior that modifies only the illumination of images while keeping other intrinsic properties unchanged. Based on this method, we can scale up the training of diffusion-based illumination editing models to large data quantities ($>$10 million), across all available data types (real light stages, rendered samples, in-the-wild synthetic augmentations, *etc.*), and using strong backbones (SDXL, Flux, *etc.*). We also demonstrate that this approach reduces uncertainties and mitigates artifacts such as mismatched materials or altered albedos.

## 1 Introduction

Editing the illumination in images is a fundamental task in deep learning and image editing. Classic computer graphics methods often model the appearance of images using physical illumination models. More recently, large diffusion-based image generators have introduced unique applications and flexible paradigms in this area, handling a wider range of "in-the-wild" lighting effects beyond simply changing the distribution of light sources, *e.g.*, generating backlighting or rim light, adding special effects like glow, glare, or the Tyndall effect, simulating shadows cast through tree shade or venetian blinds, and even manipulating human-drawn, composed, artistic, or non-photorealistic lighting conditions. These applications also provide tools for artists and designers to modify the foreground or background (*e.g.*, product images, commercial posters, *etc.*) while maintaining harmonious illumination. These illumination editing applications with generative image models hold unique industrial value for visual content creation and manipulation.

Diffusion-based illumination editing methods also present new opportunities and considerations for scaling up training and utilizing stronger backbones. Yet, training an illumination editing model at larger scales and with more diversity is more challenging than it seems. The first challenge lies in maintaining the desired model behavior to ensure proper illumination manipulation rather than deviating into unintended random behaviors. As the dataset size and diversity increase, the mapping and distribution of the learning objective can become ambiguous and uncertain. Without appropriate constraints, the training may produce a structure-guided random image generator, resulting in outputs that do not align with the desired illumination editing requirements. This deviation occurs when the model fails to learn a mapping corresponding to illumination modification, instead introducing

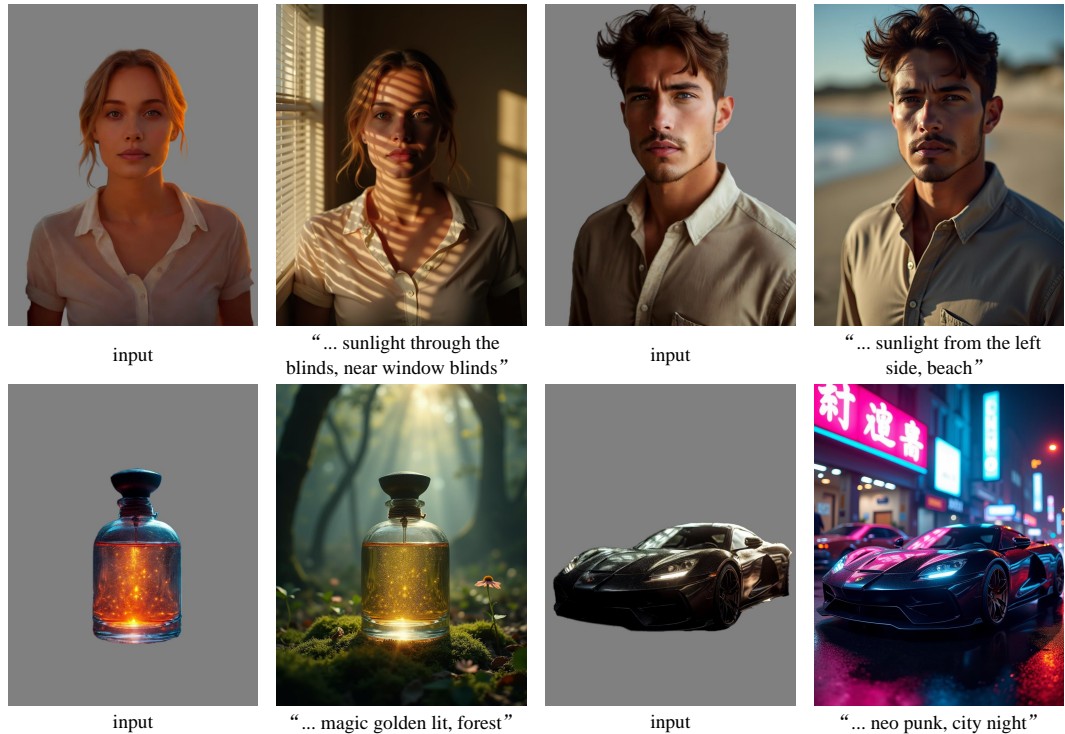

input     *"... sunlight through the blinds, near window blinds"*     input     *"... sunlight from the left side, beach"*

input     *"... magic golden lit, forest"*     input     *"... neo punk, city night"*

Figure 1: **Text-conditioned illumination modifying** with background generation (Flux.1-dev). We demonstrate the typical use case of this approach: users give an object image and illumination description, and our method generates corresponding object appearances and backgrounds.

arbitrary changes to the images, driven by dataset local minima or pretrained model default behaviors without proper alignments.

The second challenge is to preserve the underlying image details and intrinsic properties, such as albedo or reflectance colors, when modifying the illumination. Due to the stochastic nature of diffusion algorithms and the encoding-decoding processes of latent spaces, diffusion-based image generators inherently tend to introduce randomness into image contents, making it difficult to retain fine-grained details. Furthermore, effective illumination editing requires the model to have a thorough understanding of the scene to correctly adjust elements like shadows, highlights, and specular reflections. For instance, if the original image contains hard shadows, the model must first remove these existing shadows before adding new light sources and appropriate shadows. Preserving image details and intrinsic properties thus requires not only content generation but also discriminative and decomposition capabilities from the model to analyze image constituents. This necessitates careful design of training objectives and constraints to guide the learning process effectively.

In this paper, we propose a method to Impose Consistent Light (IC-Light) transport during training, grounded in the physical principle of light transport independence — the linear blending of an object's appearances under different illumination conditions is consistent with its appearance under mixed illumination. By enforcing this consistency, we introduce a strong, physically-rooted constraint that ensures the model modifies only the illumination aspects of an image while preserving other intrinsic properties such as albedo and fine image details. This approach enables stable, scalable training on over 10 million diverse samples, including real photos from light stages, rendered images, and in-the-wild images with synthetic illumination augmentations. Our method demonstrates improved precision in illumination editing, reducing uncertainties and mitigating artifacts, without altering the underlying appearance details.

This method allows us to achieve a maximized setup: expanding the dataset to over 10 million images, adopting stronger backbones like SDXL and Flux, and utilizing all available types of data sources, including real photos captured from light stages, rendered images, and in-the-wild natural or artistic

images with synthetic illumination augmentations. We provide experimental evidence to validate that increasing the training scale and diversifying data sources enhance model robustness and performance in various illumination-related downstream tasks.

Ablation experiments demonstrate that applying the IC-Light method during training improves the accuracy of illumination editing in preserving intrinsic properties like albedo and image details. Furthermore, when compared to alternative models trained on smaller or more structured datasets, our approach generalizes to a wider variety of illumination distributions, *e.g.*, rim lighting, backlighting, magic glowing, sunset halo, *etc*. We also showcase the method's ability to handle more in-the-wild illumination scenarios, including artistic and composed lighting effects. Additionally, we explore further applications, such as generating normal maps, and discuss the differences between this approach and typical mainstream geometry estimation models.

In summary, (1) we propose IC-Light, a method for scaling up the training of diffusion-based illumination editing models by imposing consistent light transport, ensuring precise illumination modification while preserving intrinsic image details; (2) we provide pretrained illumination editing models to facilitate illumination editing applications in content creation and manipulation across diverse domains; (3) we present extensive experiments to validate the scalability and performance of this approach, showing its difference from alternative methods in handling diverse illumination conditions; (4) we present additional applications, such as normal map generation and artistic lighting manipulation, further showcasing the versatility and robustness of our method in real-world, in-the-wild scenarios.

## 2 RELATED WORK

**Image Illumination Editing with Deep Neural Networks**   Learning-based methods have become important baselines in image relighting over the past decade. Sun et al. (2019) utilized deep neural networks to learn prior knowledge from light stage data. Nestmeyer et al. (2020) further enhanced the neural network's capability in relighting by modeling physical priors in the neural network training process. Pandey et al. (2021) made use of high dynamic range (HDR) lighting maps (Debevec, 2008) to train relighting models by explicitly optimizing the Phong model prior. Various baselines have been proposed to improve the efficiency, performance, and rationale of illumination modeling (Zhou et al., 2019; Sengupta et al., 2021; Hou et al., 2021; 2022; Wang et al., 2023b; Zhou et al., 2023). Switchlight (Kim et al., 2024) is a state-of-the-art relighting method trained with physically co-designed neural networks for foreground relighting. 3D facial modeling (Shu et al., 2017) or intrinsic images (Barron & Malik, 2014; Sengupta et al., 2018) have also been demonstrated to be effective in portrait relighting. Illumination stylization can also facilitate portrait relighting (Shih et al., 2014).

**Diffusion Models for Appearance and Illumination Manipulation**   Recently, due to the development of text-to-image diffusion models (Dhariwal & Nichol, 2021; Ho et al., 2020; Sohl-Dickstein et al., 2015; Song et al., 2021), a wide range of tasks in image processing have seen significant advancements (Ho & Salimans, 2021; Ho et al., 2022; Kawar et al., 2022a;b; Ramesh et al., 2022; Rombach et al., 2022; Saharia et al., 2022; Zhang et al., 2023a; Wang et al., 2022). More specifically, image editing (Alaluf et al., 2023; Brack et al., 2023; Brooks et al., 2023; Cao et al., 2023; Fu et al., 2023; Han et al., 2024; Couairon et al., 2022; Hertz et al., 2022; Meng et al., 2021; Mokady et al., 2022; Song et al., 2023; 2024; Tumanyan et al., 2023; Miyake et al., 2023; Zhang et al., 2023b; Wu & De la Torre, 2023; Huberman-Spiegelglas et al., 2023; Wallace et al., 2022) and paired image-to-image translation (Kwon & Ye, 2022; Nie et al., 2023; Sasaki et al., 2021; Wang et al., 2022; Zhang et al., 2023a; Zhao et al., 2022) have shown that fine-tuning pretrained diffusion models is an effective approach for manipulating the appearance of images. Text-to-image models have also been proven effective in depth estimation (Ke et al., 2024), normal estimation (Fu et al., 2024), and 3D construction (Anciukevičius et al., 2023; Chen et al., 2023a;b; Li et al., 2024; Lin et al., 2023; Liu et al., 2023a; 2024b; 2023b; Tang et al., 2023; Metzer et al., 2022; Wang et al., 2023c; Poole et al., 2022; Wang et al., 2023a; Xu et al., 2023). Relightful Harmonization (Ren et al., 2024) is a state-of-the-art approach for manipulating the illumination of image foregrounds using background conditions. More recently, DilightNet (Zeng et al., 2024), FlashTex (Deng et al., 2024), LightIt (Kocsis et al., 2024), and NeuralGaffer (Jin et al., 2024) have been proposed to manipulate object appearances mainly based on 3D rendering, NeRF representations, and synthetic data.

**Light Stage Methods**    A light stage (Debevec et al., 2000) is a facility used to capture the appearance of real-world objects under different illumination conditions. Major recent progress has been made towards efficiency in light patterns and neural representations of different illuminations (Fyffe & Debevec, 2015; Ghosh et al., 2011; Meka et al., 2019). Human portrait processing has been considered an important direction in light stage research (Hou et al., 2021; 2022; Nestmeyer et al., 2020; Pandey et al., 2021; Sun et al., 2019; 2020; Yeh et al., 2022; Zhang et al., 2021; 2020b). Wang et al. (2023b) pointed out that the sun can also be used as a special type of light stage. Calian et al. (2018) proposed modeling "light probes" for facial relighting. Sengupta et al. (2021) discussed that some real-world light sources can also be viewed as a light stage, like different images on desk monitor devices. Sevastopolsky et al. (2020) is another attempt to build an easier facility for more effective capturing of the light stage using a smartphone camera. Various illumination models (Debevec et al., 2000; Dorsey et al., 1995; Wenger et al., 2005) also depend on light stage measurements.

**Intrinsic Images**    Intrinsic imaging and decomposition is a long standing problem in image processing. Early approaches are often closely connected to image smoothing research like, *e.g.*, L0 (Xu et al., 2011), L1 (Bi et al., 2015), RTV (Xu et al., 2012), WLS (Farbman et al., 2008), EAP (Zhang et al., 2020a), *etc*. With datasets like Intrinsic Images in the Wild (IIW (Bell et al., 2014)) and MPI-Sintel (Butler et al., 2012), learning-based approaches are adopted. Narihira et al. (2015) trains a linear classifier contextual cues from local image patches. Zhou et al. (2015) employs a multi-stream network using both local and global inputs. Zoran et al. (2015) leverages local and global contextual information for pairwise classification. Nestmeyer & Gehler (2017) predicts a dense reflectance layer via a CNN. Narihira & Yu (2015) and Eigen & Fergus (2015) train CNNs to directly predict albedo and shading. Kim et al. (2016) uses a joint model to learn depth maps and intrinsic images with coupled activations. Shi et al. (2017) trains an encoder-decoder CNN to predict intrinsic layers using synthetic images. Barron & Malik (2012; 2020) presents a computational approach for estimating environment light from shapes. Fan et al. (2018) presents a learning-based intrinsic decomposition method that can make use of both image-space supervision and sparse supervision. Kocsis et al. (2023) propose to formulate the task of single view appearance decomposition as a probabilistic problem, and uses diffusion model to give multiple possible decomposition. IntrinsicAnything (Xi et al., 2024) is a more recent work to estimate intrinsic images using pretrained diffusion models. IntrinsicDiffusion (Luo et al., 2024) fine-tune diffusion models to jointly predict multiple intrinsic modalities (albedo, illumination, and surface geometry) from input images.

## 3    METHOD

### 3.1    IN-THE-WILD DATA DISTRIBUTION OF ILLUMINATION

As shown in Fig. 2, we model the distribution of illumination effects with multiple available types of data sources: arbitrary images, 3d data, and light stage images. These distributions allow capture of diverse and complex real-world lighting scenarios, *e.g.*, back light, rim light, glow, *etc*. For simplicity, we process all data to a common format. Every appearance image $I_L \in \mathbb{R}^{h \times w \times 3}$ is paired with a 32px environment map $L \in \mathbb{R}^{32 \times 32 \times 3}$, a foreground mask $M \in \mathbb{R}^{h \times w}$, an optional background image $B \in \mathbb{R}^{h \times w \times 3}$, and an optional degradation image $I_d \in \mathbb{R}^{h \times w \times 3}$.

**In-the-wild image augmentation**    We use data augmentation to convert an arbitrary image into paired illumination training data of images with same intrinsics (*e.g.*, albedo) but different illumination appearances. Each sample includes one appearance for input condition, another appearance for output objective, and other meta data like environment maps. The images for output objective are high-quality in-the-wild images, whereas the images for input conditions contain randomized augmentations and degradations to enhance the robustness and generalization of the diffusion model.

Specifically, we first extract environment maps for an arbitrary image by randomly choosing between two methods: We either use the method of Phongthawee et al. (2023) or a custom environment-from-normal method detailed in the supplementary material. We detect foreground mask with Zheng et al. (2024) and generate background images with a distill-accelerated (Luo et al., 2023) Stable Diffusion inpainting model. We detect prompts using the method of Xiao et al. (2023) or by using existing image prompts if the image is from text-image datasets. We then generate a "degradation appearance" that shares the same intrinsic albedo as the original image, but has completely altered illuminations;

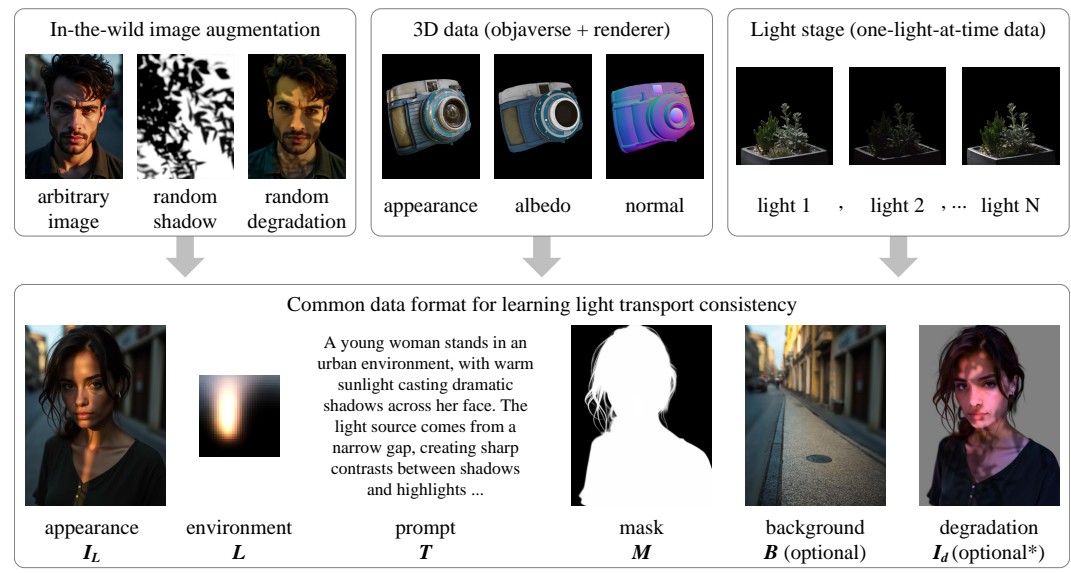

Figure 2: **Dataset collection.** We visualize various sources of the dataset and the components used during the training. Data from multiple sources are unified into a common format for neural network training. * Only "in-the-wild data augmentation" part has degradation images.

specifically we extract image albedo by randomly applying 6 albedo extraction methods in the supplementary materials. Then, we synthesize soft shading images using 3 random normal estimation methods, and synthesiz hard shadows with random shadow materials. Finally, we add a random level of specular reflection to random areas. See also the supplemental materials for full details. The shading images are 20k high-quality shadow materials purchased from several online image stocks, and 500k generated materials using a Flux LoRA trained on those 20k purchased samples. We filtered 50M images to finalize 6M images by comparing the CLIP Vision similarity to key words "beautiful lighting", "light", and "illumination" to remove images unrelated to illumination.

**3D rendering data** We render Objaverse (Deitke et al., 2023) using a method similar to G-buffer Objaverse (Zuo et al., 2024). The difference is that we use an image-based rendering pipeline written in PyTorch for faster speed. We use random environment maps obtained from previous "in-the-wild image augmentation", and use the method of Xiao et al. (2023) to detect prompts. We do not generate degradation images but directly use random unpaired environment map to render an altered appearance as $I_d$. The scale of this portion of our dataset is finalized at 4M images.

**Light Stage** We use multiple light stage datasets from Mnichelson (2006), Liu et al. (2024a), and an internal dataset with 20k light stage appearances. We pre-render all One-Light-At-a-Time (OLAT) data into the same aforementioned format. We use random environment maps obtained from previous "in-the-wild image augmentation", use the method of Xiao et al. (2023) to detect prompts, and use random unpaired environment map to render altered appearances as $I_d$.

## 3.2 Imposing Consistent Light Transport

Our goal is to learn a robust and generalized model to handle in-the-wild illumination patterns. Nevertheless, learning the large-scale, complicated, and noisy data is challenging. Without well-suited regularization and constraints, the model can easily degrade to random behaviors that do not correspond to the intended illumination editing. Our solution is to Impose Consistent Light (IC-Light) transport during training, rooted in the physical principle that the linear blending of an object's appearances under different illumination conditions is consistent with its appearance under a mixed illumination condition.

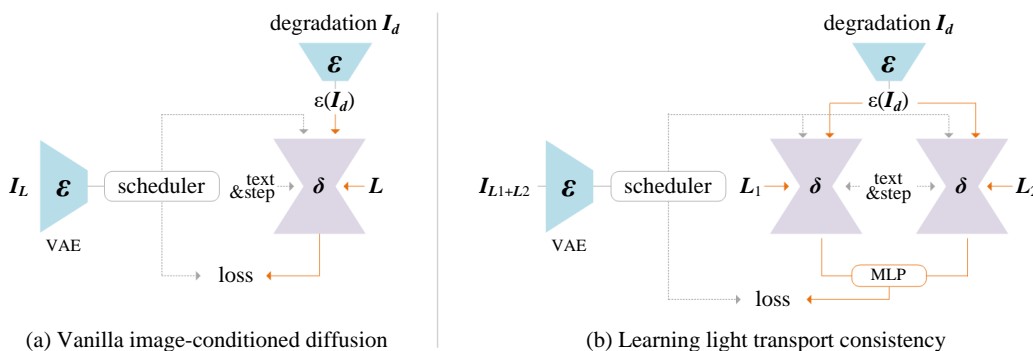

Figure 3: **Learning objective.** We visualize the learning objective of a vanilla image-conditioned diffusion model for illumination learning, and the learning objective for light transport consistency. The VAEs are frozen. Although this architecture is typical for UNet-based diffusion models, the same method also applies for (latent) diffusion transformers.

**Vanilla objective**   We start with a vanilla image-conditioned diffusion model to learn the illumination without special constraints. As shown in Fig. 3-(a), taking a typical Stable Diffusion UNet as an example, we manipulate the UNet architecture to add 4 channels to the input convolution layer to receive $I_d$, the randomly relighted appearance of the target object or the degradation image. We reshape any $32 \times 32 \times 3$ HDRI environment light source image $L$ to 3072 numbers and train a MLP from scratch with projection $3072 \rightarrow 4096 \rightarrow 4096 \rightarrow 4096 \rightarrow 2304$ (activated by Leaky RELU) and reshape the 2304 output numbers into a $3 \times 768$ embedding that can be directly received as a prompt embedding input with 3 tokens and 768 channels by SD 1.5. Given the target relighted image $I_L$, latent diffusion algorithms first encode $I_L$ as a latent image $\varepsilon(I_L)$, and then progressively add noise to the latent image to produce a noisy latent $\varepsilon(I_L)_t$, where $t$ represents the number of times noise is added. Given the set of conditions including time step $t$, illumination condition $L$, as well as the input degradation $I_d$, image diffusion algorithms learn the network $\delta$ to predict the noise with

$$\mathcal{L}_{\text{vanilla}} = \|\epsilon - \delta(\varepsilon(I_L)_t, t, L, \varepsilon(I_d))\|_2^2, \tag{1}$$

where $\epsilon$ is a diffusion target (noise or v-target for eps/v-prediction model, or flow target for flow match); $\mathcal{L}_{\text{vanilla}}$ is the cost function. Learning this objective allows for basic image relighting functionality with diffusion models. Besides, to train background-conditioned model, we concatenate $B$ to $I_d$ (and fill the extra channel with all zeros if some part of the dataset do not have backgrounds). Nevertheless, since the illumination data is challenging and noisy, this single objective will often lead to random model behaviors, *e.g.*, color mismatch, incorrect details, *etc*.

**Light transport consistency**   In computational photography, light transport theory demonstrates that, considering arbitrary appearance $I_L^*$ and the correlated environment illumination $L$, a matrix $T$ always exists so that

$$I_L^* = TL, \tag{2}$$

where $T$ can be seen as $T \in \mathbb{R}^{(h \times w \times 3) \times (32 \times 32 \times 3)}$ in our data format. The "$*$" indicates images in raw high-dynamic range. Real-world measurements (Debevec et al., 2000) validate that $T$ can always be represented with a single matrix, without any non-linear transforms. Because of this linearity, light transport explains appearance merging that

$$I_{L_1+L_2}^* = T(L_1 + L_2) = I_{L_1}^* + I_{L_2}^*, \tag{3}$$

where $L_1, L_2$ are two arbitrary environment illumination maps. This intuitively shows that the mixture of an object's appearances under separate illuminations (*e.g.*, $L_1, L_2$) is equivalent to the appearance under merged illumination (*e.g.*, $I_{L_1+L_2}^*$). This phenomenon is also validated by real-world measurements, *e.g.*, (Haeberli, 1992), and we attach related validating examples in the supplementary materials.

In this paper, we observe that the appearance $I$ in Eq. (3) can be replaced by arbitrary diffusion targets thanks to its linearity. For instance, consider a simple k-diffusion epsilon target at sigma-space step $\sigma_t$, estimated noise $\epsilon_L$ (conditioned on $L$), and noisy image $I_{\sigma_t}$, the estimated clean appearance can

be written as $\hat{\boldsymbol{I}}_{\boldsymbol{L}} = (\boldsymbol{I}_{\sigma_t} - \epsilon_{\boldsymbol{L}})/\sigma_t$. By applying Eq. (3) as $\hat{\boldsymbol{I}}_{\boldsymbol{L_1+L_2}} = \hat{\boldsymbol{I}}_{\boldsymbol{L_1}} + \hat{\boldsymbol{I}}_{\boldsymbol{L_2}}$ we will also have $\lambda\epsilon_{\boldsymbol{L_1+L_2}} = \epsilon_{\boldsymbol{L_1}} + \epsilon_{\boldsymbol{L_2}}$ where $\lambda$ is a constant scaling factor (usually $\lambda = 2$, see also Appendix. A). As a result, the term in Eq. (3) can be replaced by any diffusion targets, *e.g.*, eps-prediction, v-prediction, flow match, *etc.*, as long as the target itself is linear and first-order.

The core idea of light transport consistency is to guarantee Eq. (3) during the diffusion training process so as to constrain the model to only modify image illumination without changing other intrinsic properties (*i.e.*, to keep the internal light transport $\boldsymbol{T}$ unchanged). This can be achieved by minimizing $||\boldsymbol{I}_{\boldsymbol{L_1+L_2}}^* - (\boldsymbol{I}_{\boldsymbol{L_1}}^* + \boldsymbol{I}_{\boldsymbol{L_2}}^*)||_2^2$ (where $|| \cdot ||_2^2$ is L2 norm). Using the aforementioned conversion, we can write it as a loss function for eps-prediction model $||\epsilon_{\boldsymbol{L_1+L_2}} - (\epsilon_{\boldsymbol{L_1}} + \epsilon_{\boldsymbol{L_2}})||_2^2$.

For practical implementation, considering that most diffusion models are not pixel diffusion models trained on HDR images, an conversion is needed for latent diffusion or LDR pixel diffusion. We use a simple learnable Multi-Layer Perceptron (MLP) $\phi(\cdot, \cdot)$ to learn an implicit adaptation among potentials data domains (LDR, HDR, Latent), by replacing the sum term in Eq. 3. Taking eps-prediction as an example, we have the final form of light transport consistency

$$\mathcal{L}_{\text{consistency}} = \|\boldsymbol{M} \odot (\epsilon_{\boldsymbol{L_1+L_2}} - \phi(\epsilon_{\boldsymbol{L_1}}, \epsilon_{\boldsymbol{L_2}}))\|_2^2, \tag{4}$$

where $\phi(\cdot, \cdot)$ is a 5-layer MLP with hidden state 128, and input/output same as latent channels for different models, and $\odot$ is pixel-wise multiplying with foreground mask $\boldsymbol{M}$ (resized to same size as latent images). During training, we synthesize $\boldsymbol{L}_1, \boldsymbol{L}_2$ by generating random $4 \times 4$ masks from uniform distribution and resize the masks to same shape as $\boldsymbol{L}$, then, we view the masked area as $\boldsymbol{L}_1$ and unmasked areas as $\boldsymbol{L}_2$, ensuring $\boldsymbol{L} = \boldsymbol{L}_1 + \boldsymbol{L}_2$. This loss function can be fully expanded as

$$\mathcal{L}_{\text{consistency}} = \|\boldsymbol{M} \odot (\epsilon - \phi(\boldsymbol{\delta}(\boldsymbol{\varepsilon}(\boldsymbol{I}_{\boldsymbol{L}})_t, t, \boldsymbol{L}_1, \boldsymbol{\varepsilon}(\boldsymbol{I}_d))), \boldsymbol{\delta}(\boldsymbol{\varepsilon}(\boldsymbol{I}_{\boldsymbol{L}})_t, t, \boldsymbol{L}_2, \boldsymbol{\varepsilon}(\boldsymbol{I}_d)))\|_2^2, \tag{5}$$

where each component is visualized in Fig 3-(b).

**Joint learning objective**   The final learning objective can be written as

$$\mathcal{L} = \lambda_{\text{vanilla}}\mathcal{L}_{\text{vanilla}} + \lambda_{\text{consistency}}\mathcal{L}_{\text{consistency}}, \tag{6}$$

where $\mathcal{L}$ is the merged objective, and we use $\lambda_{\text{vanilla}} = 1.0, \lambda_{\text{consistency}} = 0.1$ as default weights.

# 4 EXPERIMENT

## 4.1 EXPERIMENTAL DETAILS

We use the AdamW optimizer with the learning rate of 1e-5 for training the entire framework. The pretrained Stable Diffusion models used are SD 1.5, SDXL, and Flux.1.0-dev. The training was conducted on 8 H100 80GB NVLink GPUs. We select the largest possible batch size for each model. For the SD 1.5 version of the model, the training took 100 hours. For the SDXL model, we first trained at 512 resolution for 80 hours, then fine-tuned at 1024 resolution for 60 hours.

The training process for the Flux model was more complex due to the large size of the Flux model. We adopted a multi-stage training strategy, where we separately trained the double-stream and single-stream parts of the model, using gradient freezing to freeze some parts of the gradient graph. This allowed us to train in larger batch sizes. More training details can be found in the supplementary material.

We use scheduled probability to balance the multiple training datasets, the in-the-wild image data and 3D rendering data appeared with equal probability during the initial stage of training. As training iterations increased, the probability of light stage data appearing in each batch increases. This allowed us to leverage a smaller portion of high-quality light stage data to improve the final model performance. In the beginning stages of training, the probabilities of in-the-wild data and 3D data were both 0.5, with light stage being 0.0. After 100,000 iterations, the probabilities were adjusted to 0.35 for in-the-wild data, 0.35 for 3D data, and 0.3 for light stage data. These probabilities were adjusted linearly throughout the training process.

## 4.2 ABLATIVE STUDY

We conducted an ablative study to understand the importance of different components. We first resume the model from training but removing the in-the-wild image augmentation data. As seen in

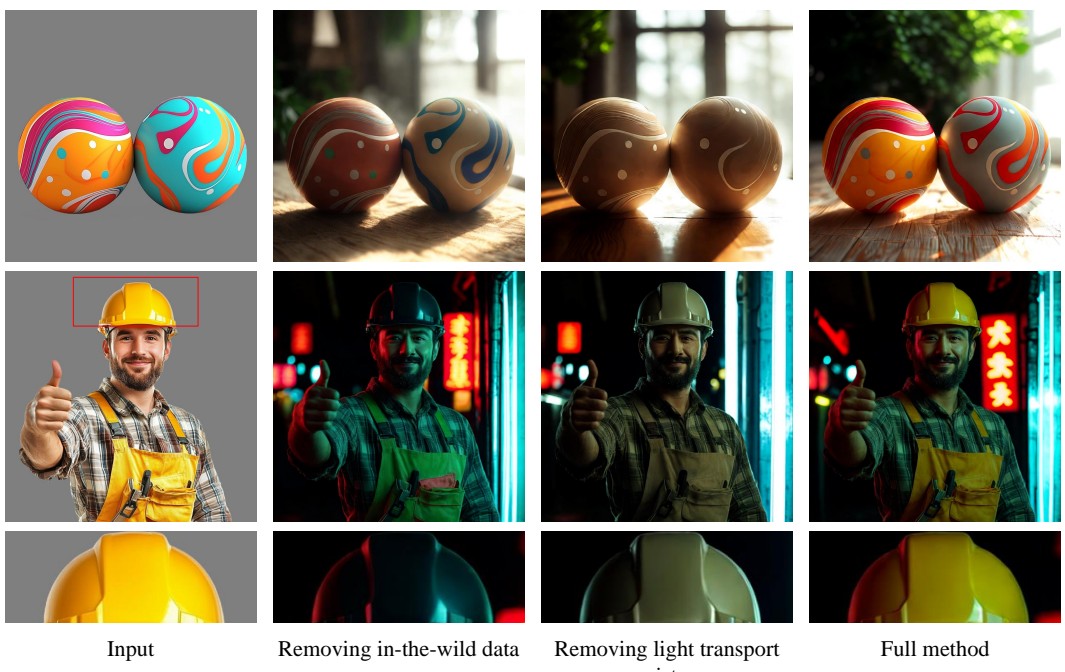

| Input | Removing in-the-wild data | Removing light transport consistency | Full method |

Figure 4: **Ablative Study.** We present results by removing the light transport consistency or in the wild data. More results are in the supplementary material. Results in this figure are from Stable Diffusion 1.5 version of our model. Prompts are "toy in room, studio lighting", and "a handsome man, neon city".

Fig. 4, removing in-the-wild data severely impacted the model's generalization capability, especially for complex images like portraits. For example, hats on portraits that weren't present in the training data would often be rendered in incorrect colors (e.g., changing from yellow to black).

We also experimented with removing the light transport consistency. Without this constraint, the model's ability to generate consistent illumination and retain intrinsic properties such as albedo (reflectance color) significantly decreased. For example, the red and blue differences vanished in some images, and noticeable issues with color saturation are observed in the output.

The full method, which combines multiple data sources and enforces light transport consistency, produces a well-balanced model capable of generalizing across a range of scenarios. It also retains fine-grained image details and intrinsic properties, such as albedo, while reducing errors in output images. More examples are in the supplementary materials.

### 4.3 ADDITIONAL APPLICATIONS

As shown in the Fig. 5, we demonstrate additional applications such as using background conditions for illumination harmonization. Trained on extra channels for background conditions, our model can perform illumination generation conditioned solely on a background image without relying on environment maps. Additionally, our model supports different base models such as SD1.5, SDXL, and Flux, and the capabilities of these models are reflected in the generated results.

Multiple inferences from our method generate consistent appearances, which can be blended into normal maps. Specifically, for each object, we treat the global average of all relighted appearances as the albedo (diffuse color) $A$ and divide the independent relighted appearances $I_{L_i}$ by the albedo to obtain shading maps $S_{L_i}$ as $S_{L_i} = \frac{I_{L_i}}{A}$, where $L_i$ is the $i$-th light source. The two vertical shading maps $S_{L_{\text{up}}}$ and $S_{L_{\text{down}}}$ are averaged to form the green channel of the normal map $N$, and the two horizontal shading maps $S_{L_{\text{left}}}$ and $S_{L_{\text{right}}}$ are averaged to form the red channel of the normal map as $N_{\text{green}} = \frac{S_{L_{\text{up}}} - S_{L_{\text{down}}}}{2}, N_{\text{red}} = \frac{S_{L_{\text{left}}} - S_{L_{\text{right}}}}{2}$. Finally, the blue channel is padded to ensure the normal

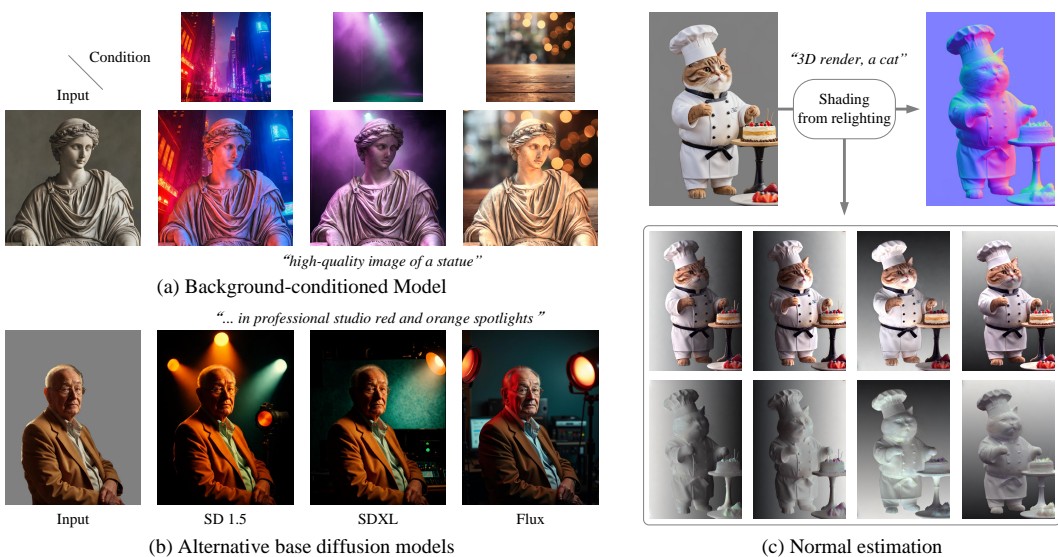

(a) Background-conditioned Model

(b) Alternative base diffusion models

(c) Normal estimation

Figure 5: **Additional applications.** We show that this model supports more types of inputs like background conditions, and various base models like SD1.5/SDXL/Flux. We also show that multiple inferences of this method yields consistent appearances that can be merged into normal maps.

map $N$ is a unit vector for each pixel as $N_{\text{blue}} = \sqrt{1 - N_{\text{red}}^2 - N_{\text{green}}^2}$ and this process generates a complete normal map that can be used for further rendering. We visualize examples in Fig. 5-(c) by transforming each channel by $\frac{N_{...}+1}{2}$. We also point out that this normal extraction is an empirical method since the neural models are not optimized to approximate light stage ground truths or 3D normal maps. The extraction of normal maps relies completely on the model's capability to produce consistent appearances in different illumination conditions.

## 4.4 QUANTITATIVE EVALUATION

We conducted quantitative comparisons using metrics such as Peak Signal-to-Noise Ratio (PSNR), Structural Similarity Index (SSIM), and Learned Perceptual Image Patch Similarity (LPIPS). We extracted a subset of 50,000 unseen 3D rendering data samples from the dataset for evaluation, ensuring that the model had not encountered these samples during training.

The tested methods are SwitchLight (Kim et al., 2024), DiLightNet (Zeng et al., 2024), and variants of our method without certain components (e.g., without light transport consistency, without augmentation data, without 3D data, and without light stage data). As shown in Table 1, our method outperforms others in terms of LPIPS, indicating superior perceptual quality. Models trained only on 3D data achieved the highest PSNR, but this is likely due to a evaluation bias towards the rendering data (since this test only use 3D rendering data). The full method, which combines multiple data sources, achieved a balance between perceptual quality and performance.

Table 1: Quantitative tests of ablative architectures and alternative methods.

| Method | PSNR ↑ | SSIM ↑ | LPIPS ↓ |
|---|---|---|---|
| SwitchLight | 18.45 | 0.7024 | 0.3245 |
| DiLightNet | 21.78 | 0.8013 | 0.1721 |
| w/o LTC | 20.32 | 0.7542 | 0.1927 |
| w/o aug. data | 23.95 | 0.8723 | 0.1115 |
| w/o 3d data | 22.10 | 0.8041 | 0.1298 |
| w/o light stage | 23.70 | 0.8501 | 0.1077 |
| Ours | 23.72 | 0.8513 | 0.1025 |

## 4.5 VISUAL COMPARISON

We also conducted visual comparisons with previous methods. As seen in Fig. 6, compared to Relightful Harmonization (Ren et al., 2024), this model demonstrates higher robustness to shadows, thanks to a larger and more diverse training dataset. SwitchLight (Kim et al., 2024) and this model produces competitive relighting results. The quality of the normal maps of this method is a bit more detailed due to the methods of merging and deriving shadow from multiple appearances. Addition-

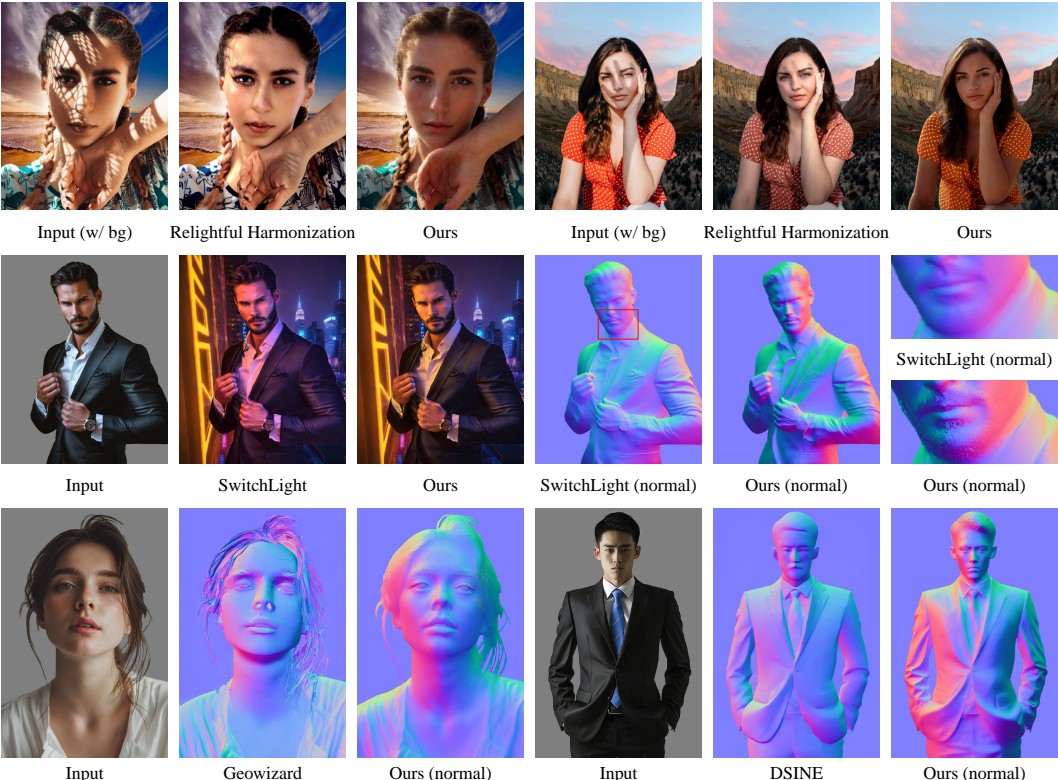

Figure 6: **Visual Comparison.** The Relightful Harmonization results are taken from official paper Ren et al. (2024) while other methods use official code bases or services.

ally, the normal maps produced by this model exhibit higher quality for human than alternatives GeoWizard (Fu et al., 2024) and DSINE (Bae & Davison, 2024).

## 5 CONCLUSION

In this paper, we propose an approach for scaling up the training of diffusion-based illumination editing models by imposing consistent light transport (IC-Light). Our method ensures desired illumination manipulation while preserving intrinsic image properties, such as albedo and fine details. This is achieved through light transport consistency, a physically grounded constraint that helps stabilize training across diverse data sources, including in-the-wild images, 3D rendered data, and light stage captures. We demonstrate through extensive experiments and ablation studies that this approach reduces uncertainties, prevents artifacts, and improves model generalization to various illumination conditions. Additionally, our method supports a range of applications, such as background-aware relighting and normal map generation, and it scales to large datasets and strong model backbones like SDXL and Flux. The results validate a robust method well-suited for industrial and creative applications in image-based illumination editing.

### ACKNOWLEDGMENTS

This work was partially supported by the Brown Institute for Media Innovation and by Google through their affiliation with Stanford Institute for Human-centered Artificial Intelligence (HAI).

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

## A  SCALING FACTOR IN DIFFUSION OBJECTIVES

We observed that certain diffusion objectives, such as $\epsilon$-prediction, introduce an extra scaling factor $\lambda$ such that

$$\lambda I_{L_1+L_2} = I_{L_1} + I_{L_2}, \tag{7}$$

where $\lambda$ corresponds to the number of blended elements (e.g., $\lambda = 2$ in this case). However, other diffusion objectives, such as $x_0$-prediction, remain unaffected. Similar to the HDR/LDR scaling, this scaling is implicitly learned by the MLP.

For example, when blending $\lambda = 2$ objectives:

$$\lambda I_{L_1+L_2} = I_{L_1} + I_{L_2}, \tag{8}$$

we have:

$$\lambda \frac{I_{\sigma_t}^L - \epsilon_{L_1+L_2}}{\sigma_t} = \frac{I_{\sigma_t}^L - \epsilon_{L_1}}{\sigma_t} + \frac{I_{\sigma_t}^L - \epsilon_{L_2}}{\sigma_t}, \tag{9}$$

$$\lambda I_{\sigma_t}^L - \lambda \epsilon_{L_1+L_2} = I_{\sigma_t}^L - \epsilon_{L_1} + I_{\sigma_t}^L - \epsilon_{L_2}, \tag{10}$$

$$\lambda \epsilon_{L_1+L_2} = \epsilon_{L_1} + \epsilon_{L_2}. \tag{11}$$

Thus, the MLP implicitly scales the data by a factor of $1/\lambda$. Notably, whether we use $\{I_{\sigma_t}^L, I_{\sigma_t}^{L_1}, I_{\sigma_t}^{L_2}\}$ or not makes no difference. Even in ideal cases where all these values are available and applied, with ground truth noise $\epsilon$ and $\lambda = 2$, we still obtain:

$$\lambda I_{L_1+L_2} = I_{L_1} + I_{L_2}, \tag{12}$$

leading to:

$$\lambda \frac{I_{\sigma_t}^{L_1+L_2} - \epsilon_{L_1+L_2}}{\sigma_t} = \frac{I_{\sigma_t}^{L_1} - \epsilon_{L_1}}{\sigma_t} + \frac{I_{\sigma_t}^{L_2} - \epsilon_{L_2}}{\sigma_t}, \tag{13}$$

$$\lambda I_{\sigma_t}^{L_1+L_2} - \lambda \epsilon_{L_1+L_2} = I_{\sigma_t}^{L_1} - \epsilon_{L_1} + I_{\sigma_t}^{L_2} - \epsilon_{L_2}, \tag{14}$$

$$\lambda (I_{L_1+L_2} + \sigma_t \epsilon) - \lambda \epsilon_{L_1+L_2} = (I_{L_1} + \sigma_t \epsilon) + (I_{L_2} + \sigma_t \epsilon) - (\epsilon_{L_1} + \epsilon_{L_2}), \tag{15}$$

$$I_{L_1} + I_{L_2} + 2\sigma_t \epsilon - \lambda \epsilon_{L_1+L_2} = I_{L_1} + \sigma_t \epsilon + I_{L_2} + \sigma_t \epsilon - (\epsilon_{L_1} + \epsilon_{L_2}), \tag{16}$$

$$\lambda \epsilon_{L_1+L_2} = \epsilon_{L_1} + \epsilon_{L_2}. \tag{17}$$

This confirms that using $I_{\sigma_t}^L$ or $\{I_{\sigma_t}^{L_1}, I_{\sigma_t}^{L_2}\}$ does not affect the final result, and the MLP implicitly learns to scale the data by $1/\lambda$. Notably, this scaling effect is only observed in certain objectives such as $\epsilon$-prediction, while other objectives like $x_0$-prediction remain unaffected.

