


Scene with illumination A
(real photo)

Scene with illumination B
(real photo)

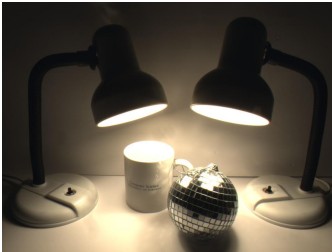
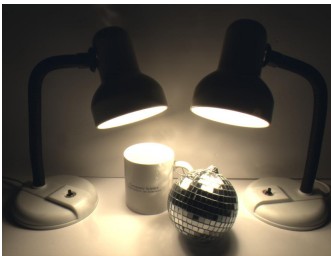
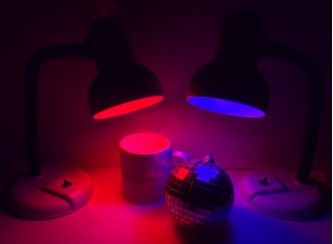

Scene with illumination C
(real photo)

Blending of real photo A and B
(computed image)

Altered Blending with color tone
(computed image)

Figure 1: Examples for "the linear blending of an object's appearances under different illumination conditions is consistent with its appearance under mixed illumination". Images from OToole (2016).

# 1 LIGHT TRANSPORT CONSISTENCY EXAMPLE EVIDENCES

We attach additional example (OToole, 2016) as a further validation and explanation for the root of the "light transport consistency" in the main paper: the linear blending of an object's appearances under different illumination conditions is consistent with its appearance under mixed illumination.

As shown in Fig.1, the first row presents the scene's two appearances under two different illumination conditions (one with left light A and another with right light B). The linear blending of the two appearances yields an computed image, which is consistent with the scene's real photo appearance if both lights are turned on. This consistency is physically precise in high-dynamic range color space. This phenomenon is equivalent to main paper's Eq. (3).

This blending can also be used in creating extra appearances by merging with channel weights (color tones).

# 2 ADDITIONAL IMPLEMENTATION DETAILS ABOUT DATASET

**Estimation environment maps**   To estimate environment maps, we randomly choose the below two methods.

- An implementation of Phongthawee et al. (2023). Probability is 30%.
- A environment-from-normal method. Probability is 70%. See the below section.

**In-the-wild image augmentation**   To create a degradation image on an existing image, we first estimate albedo using a random method in the below methods:

- Directly using original image as albedo. Probability is 30%.
- An implementation of Careaga et al. (2023). Probability is 20%.
- An implementation of Xi et al. (2024). Probability is 10%.
- An implementation of Ye et al. (2024)'s delight method. Probability is 10%.

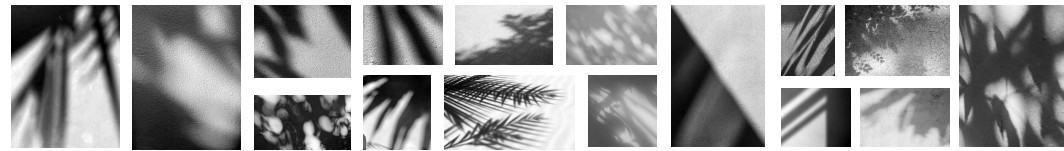

Figure 2: Examples of shadow materials. We present examples of shadow material images during training for synthesizing image degradations.

- An implementation of Bi et al. (2015). Probability is 15%.
- An implementation of Xu et al. (2011). Probability is 15%.

We then create soft shadow (with colors), we use normal maps to synthesize shadows. We use the below methods with equal probability.

- An implementation of Bae & Davison (2024).
- An implementation of Fu et al. (2024).
- An implementation of Khirodkar et al. (2024).

Those normal maps are then mapped with random environment maps (estimated in aforementioned section) to create shadow maps. Then, shadow maps are multiplied to albedos.

Note that at this step, we already have the normal map and the albedo's reflectance. We divide the original image by the albedo to obtain the a shading map. Then, the shading map is inversely projected onto the normal sphere through pixel normal directions, while maintaining the HDR (High Dynamic Range). If a position on the normal sphere has multiple values mapped, we take the average. If there are no pixels mapped to a point on the discovery sphere, we use Poisson image inpainting Pérez et al. (2003) to fill in the blank pixels on the discovery sphere. Afterward, we map the direction and color on the discovery sphere (in HDR) to an environment map.

After that, we blend hard shadows. The shadow images are 20k high-quality shadow materials purchased from several online image stocks, and 500k generated materials using a Flux LoRA trained on those 20k purchased samples. We use two random environment maps to render a same normal map to get two shading colors, then we mask the two shading maps with the shadow mask to get hard shadow.

Finally, we add random some specular reflections to random areas. We use the normal map and random environment maps to get random specular reflections. After that, we randomly sample another hard shadow image to mask some random areas. We only add specular reflections to masked areas. To avoid adding too many specular reflections, we view each specular reflection as a connected region, and randomly drop 70% such connected regions.

## 3 ADDITIONAL IMPLEMENTATION DETAILS ABOUT FLUX TRAINING SCHEDULING

Flux.1-dev has two parts: double-stream blocks, and single-stream blocks. All blocks contributes to 12 billion parameters, which is very large and difficult to train.

We view it as 3 connected parts: (1) all double-stream blocks, (2) first 50% single-stream blocks, and (3) last 50% single-stream blocks. We first train those blocks individually with a relatively large batch size, then train multiple blocks jointly with lower batch size.

Taking PyTorch as an example, if a parts is a beginning part, to turn off the gradient computation, we only need "f(x)=torch.Tensor.detach(x)" inside "torch.no_grade" to completely avoid optimization.

If a parts is an ending part, we use "y=x + torch.Tensor.detach(f(x) - x)" together with f(x) inside "torch.no_grade" to avoid gradient computation.

We first train at resolution 512 to "warm-up" the training (until seeing correlation between condition images and results) and then finetune at native resolution 1024.

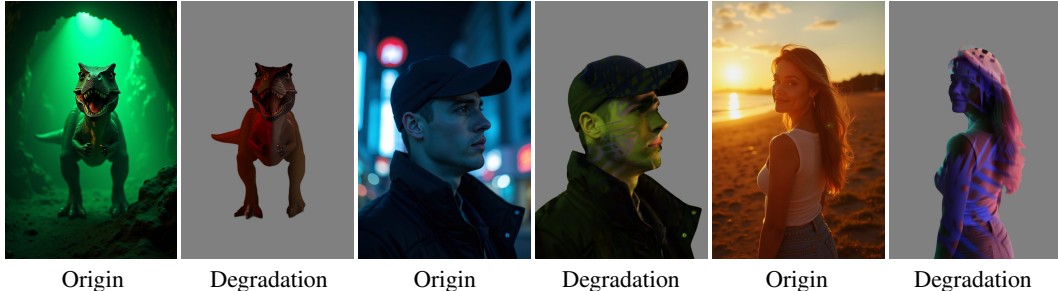

| Origin | Degradation | Origin | Degradation | Origin | Degradation |

Figure 3: Examples of illumination degradations. We present examples of synthesized data with illumination degradations.

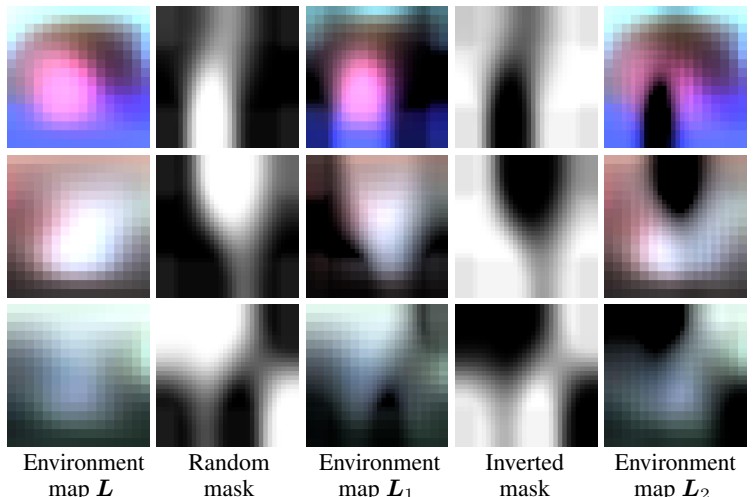

| Environment map $L$ | Random mask | Environment map $L_1$ | Inverted mask | Environment map $L_2$ |

Figure 4: Examples of decomposed environment maps. We present examples to use random masks to decompose environment map $L$ into $L_1$ and $L_2$. Note that $L = L_1 + L_2$. A typical full environment map is usually of ratio 2:1, with size $64 \times 32$ when convoluted. We use the front half (facing the image) of the convoluted environment map, which is $32 \times 32$. Using the front half makes normal-based environment extraction easier (since the image-space normals often do not have any pixels facing to the back half). Besides, the back halves of environment maps from DiffusionLight Phongthawee et al. (2023) are usually not strictly correlated to image contents and can be excluded.

## 4 ADDITIONAL DATA VISUALIZATION

We present additional examples for shadow materials in Fig. 2.

We present additional illumination degradation examples in Fig. 3.

We present additional examples for decomposing environment maps in Fig. 4.

## 5 ADDITIONAL RESULTS AND ABLATIVE STUDY

We present additional results and ablative study in Fig. 5-9.

We present additional background-conditioned results in Fig. 10-11.

We present additional examples of normal map blending in Fig. 12.

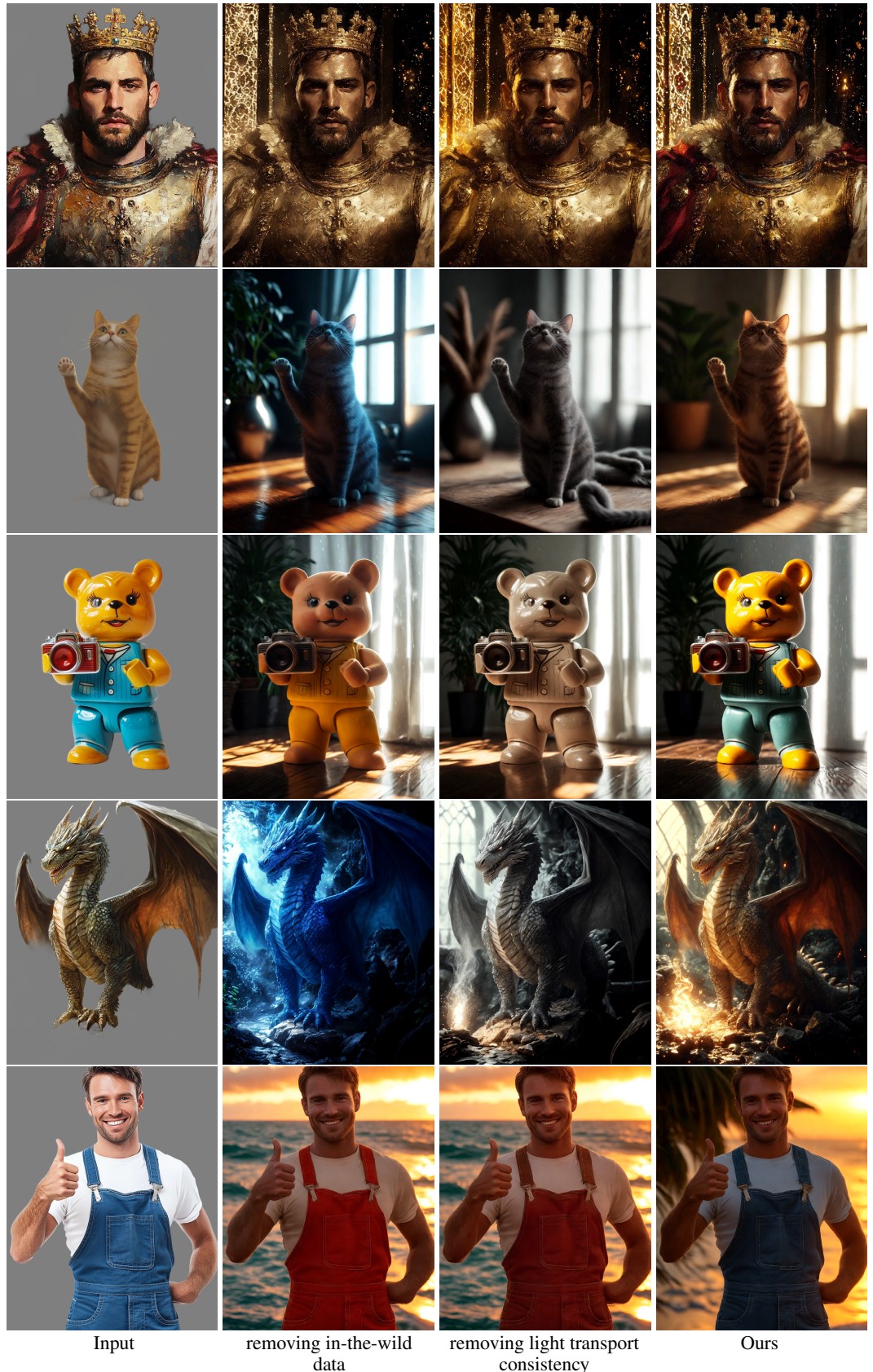

|  |  |  |  |
|---|---|---|---|
| Input | removing in-the-wild data | removing light transport consistency | Ours |

Figure 5: Additional Results and Ablative Study. Seed is always 12345. The prompts are "king, magic lit", "cat, soft studio lighting", "toy, natural lighting", "dragon, magic lit", "handsome man, detailed face, sunset over sea".

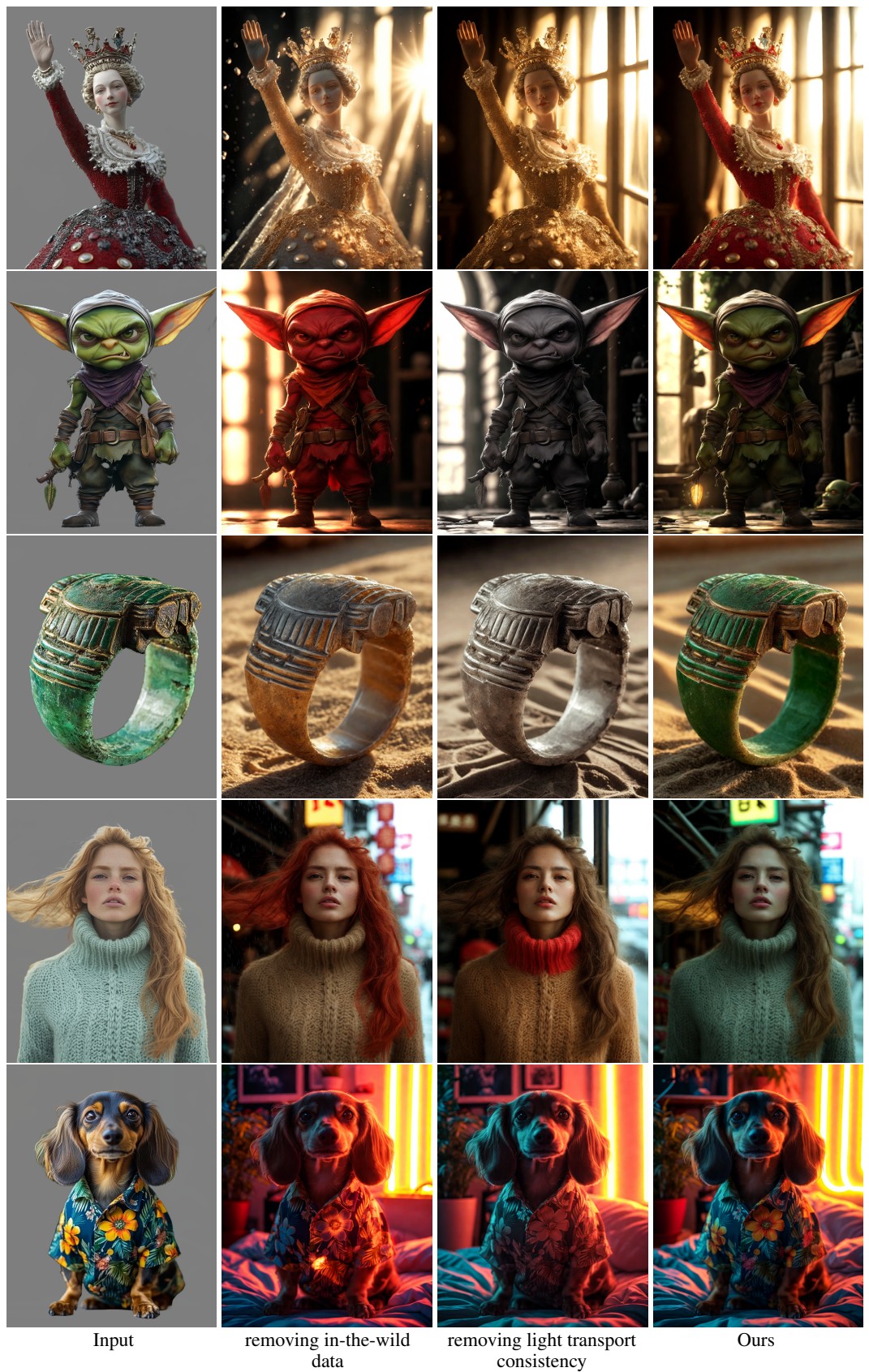

|         Input         | removing in-the-wild data | removing light transport consistency | Ours |
|:---------------------:|:-------------------------:|:------------------------------------:|:----:|

Figure 6: Additional Results and Ablative Study. Seed is always 12345. The prompts are "toy queen, natural lighting", "goblin, magic lit", "ring, on sands", "beautiful woman, detailed face, neon, Wong Kar-wai, warm", "dog, neon light, warm atmosphere, at home, bedroom".

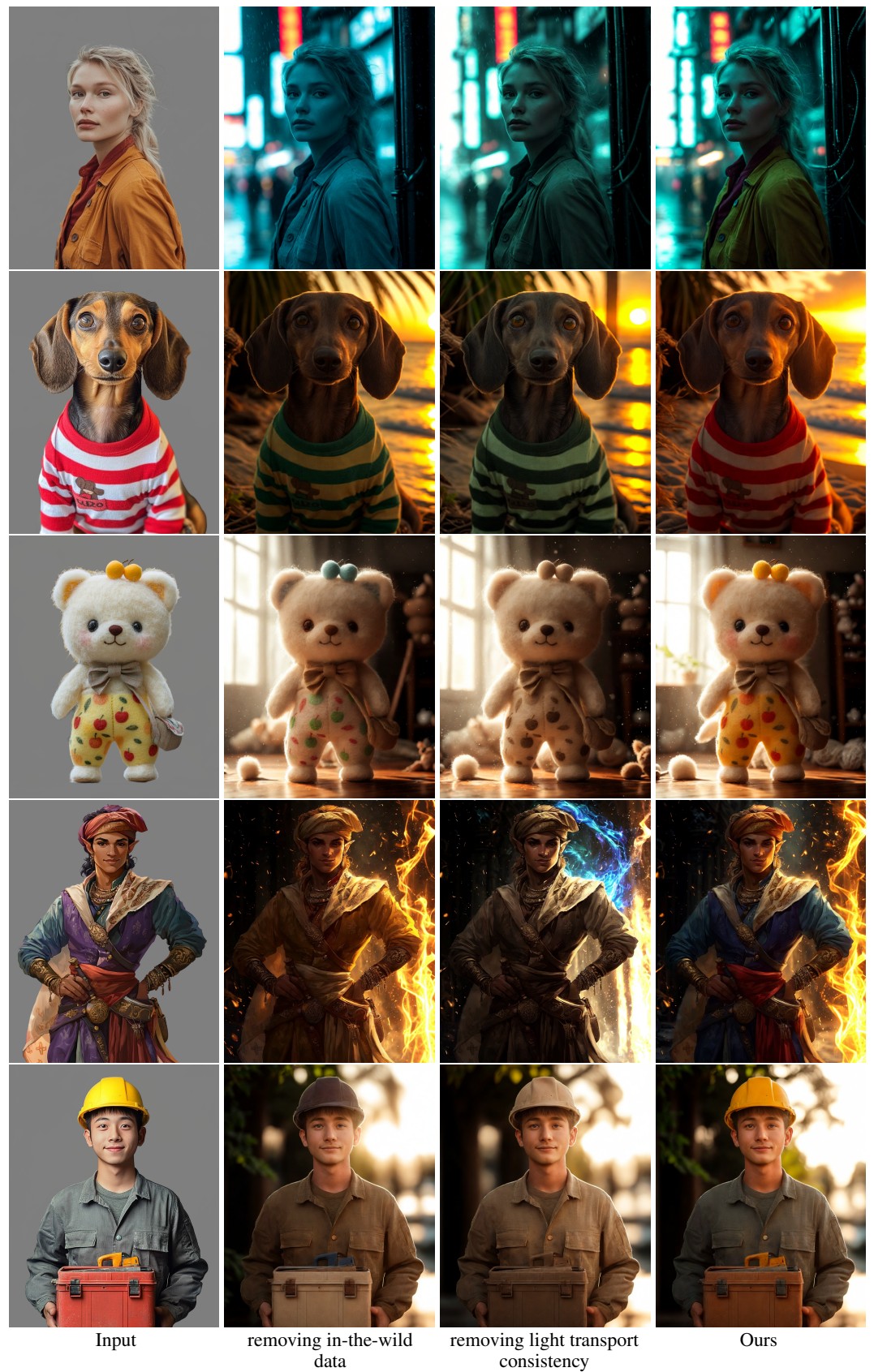

| Input | removing in-the-wild data | removing light transport consistency | Ours |

Figure 7: Additional Results and Ablative Study. Seed is always 12345. The prompts are "beautiful woman, blue, detailed face, sci-fi RGB glowing, cyberpunk", "dog, home atmosphere, sunset over sea", "toy, soft studio lighting", "character, magic lit", "handsome man, detailed face, natural lighting".

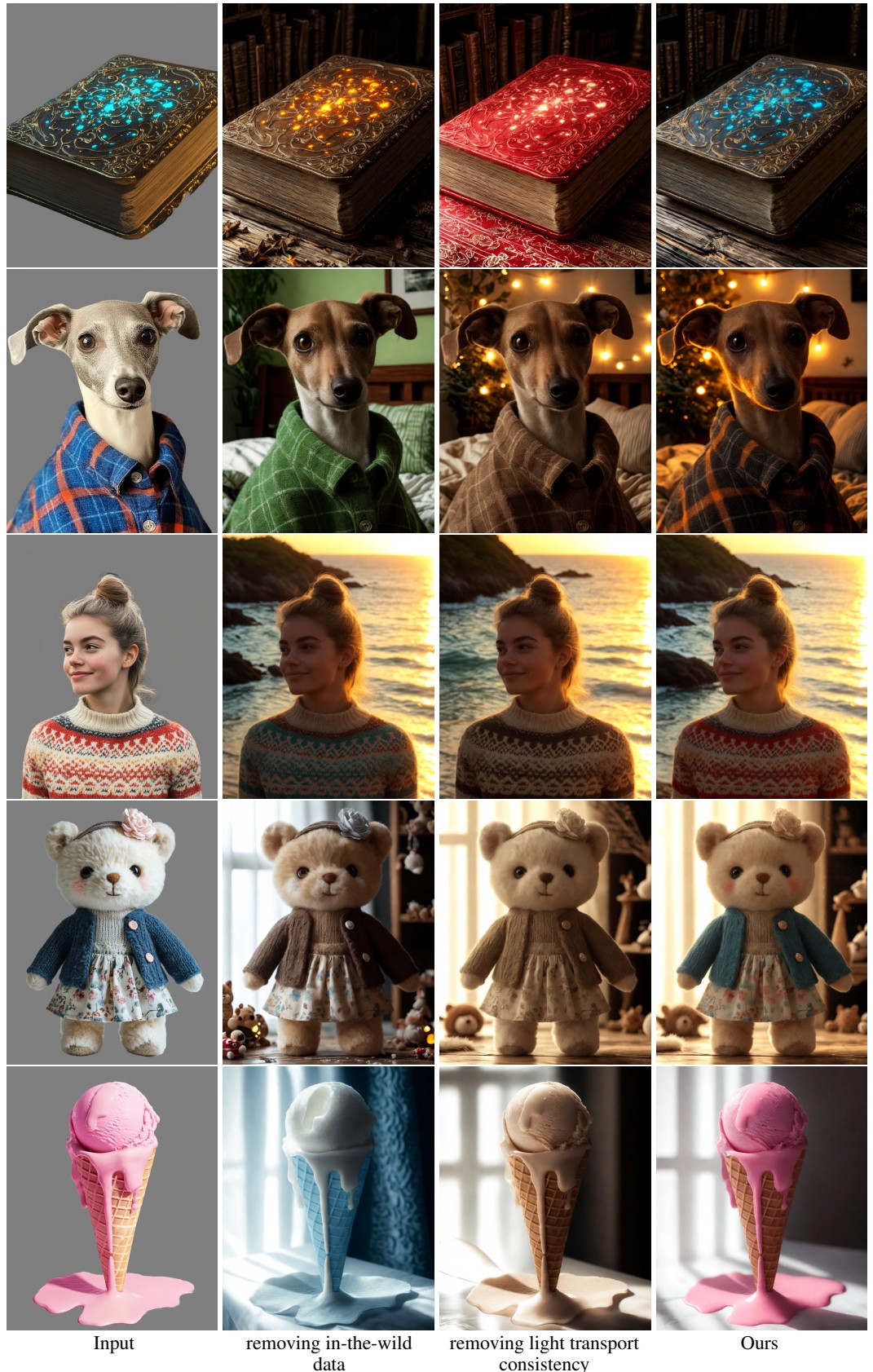

| Input | removing in-the-wild data | removing light transport consistency | Ours |

Figure 8: Additional Results and Ablative Study. Seed is always 12345. The prompts are "book, magic lit", "dog, home atmosphere, cozy bedroom illumination", "beautiful woman, detailed face, sunset over sea", "toy, magic lit", "ice cream, soft studio lighting".

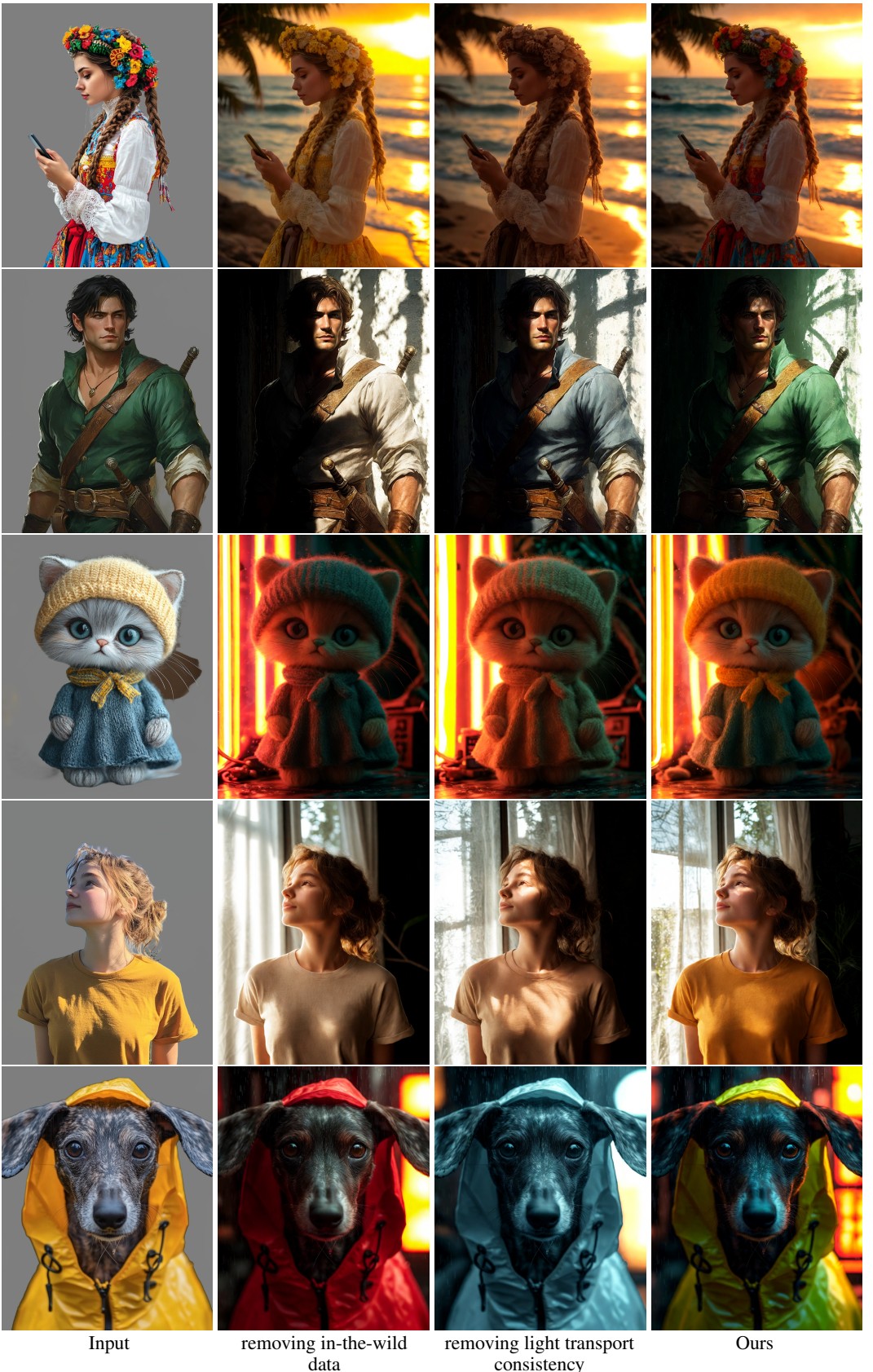

| Input | removing in-the-wild data | removing light transport consistency | Ours |

Figure 9: Additional Results and Ablative Study. Seed is always 12345. The prompts are "beautiful woman, detailed face, sunset over sea", "handsome man, shadow from window", "toy, home atmosphere, neon, Wong Kar-wai, warm", "beautiful woman, detailed face, natural lighting", "dog, sci-fi RGB glowing, cyberpunk".

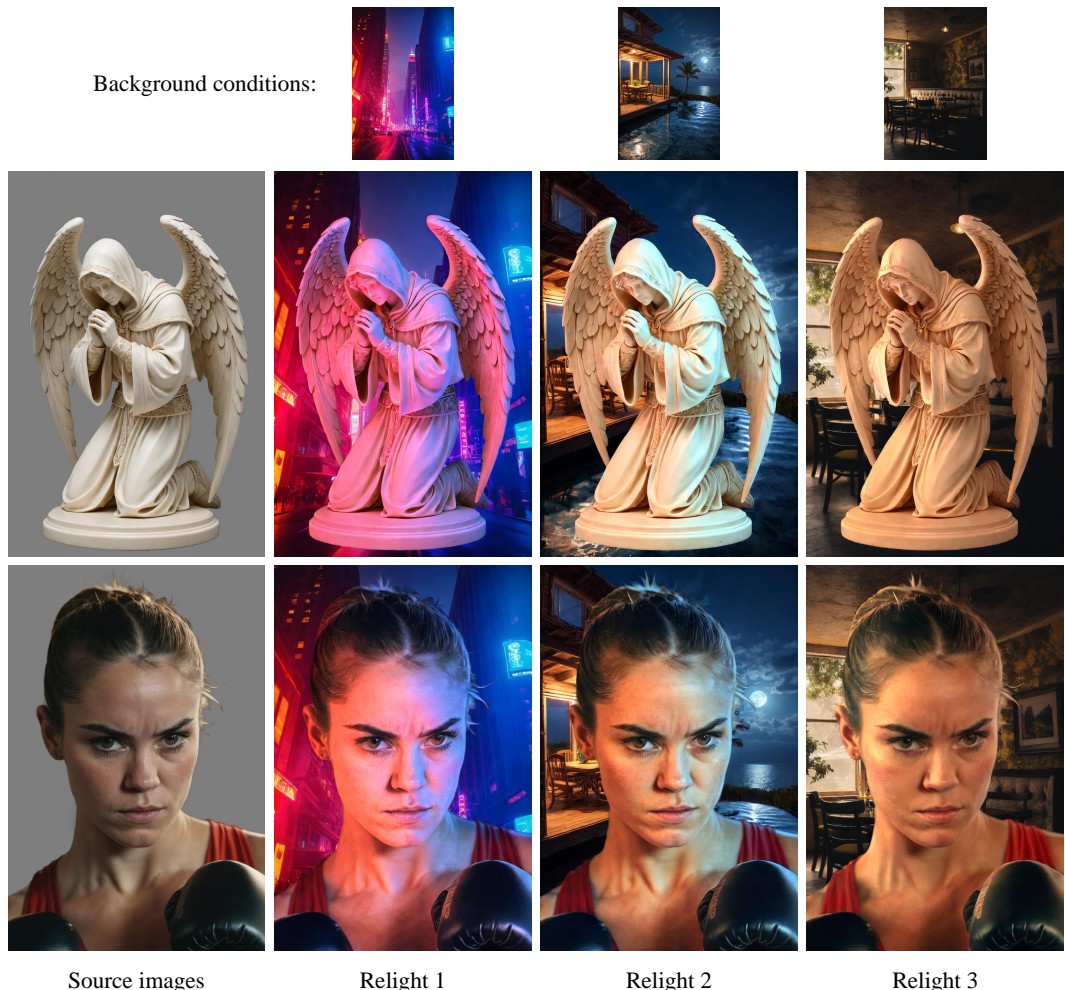

Background conditions:

Source images    Relight 1    Relight 2    Relight 3

Figure 10: **Background-conditioned Foreground Relighting.** We demonstrate the application that relights the image foregrounds using backgrounds as conditions.

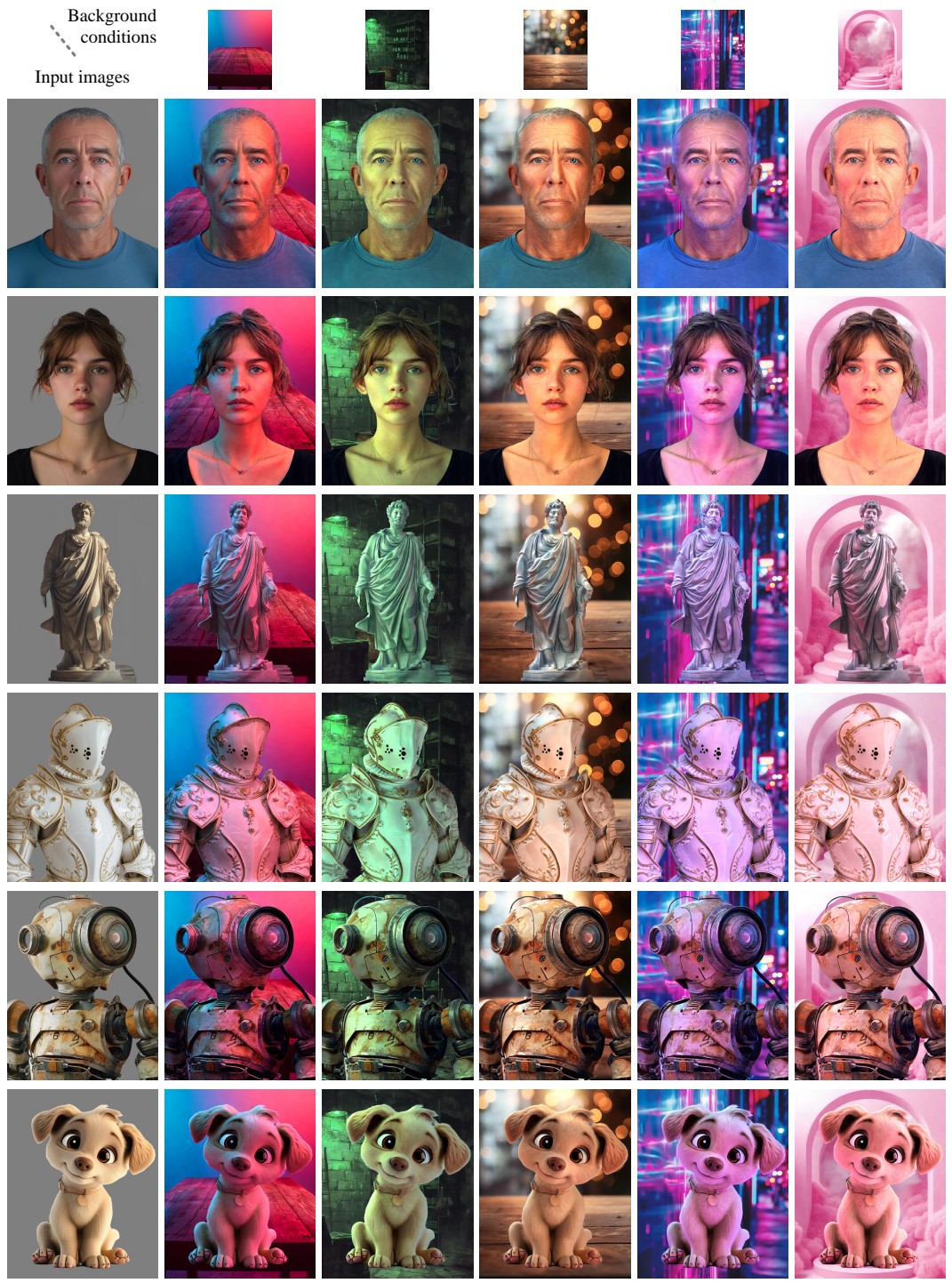

Figure 11: **Qualitative Results on Background-Conditioned Image Relighting.** Our framework relight single images using background images as conditions, without requiring light source data. The model estimate illumination in the background automatically.

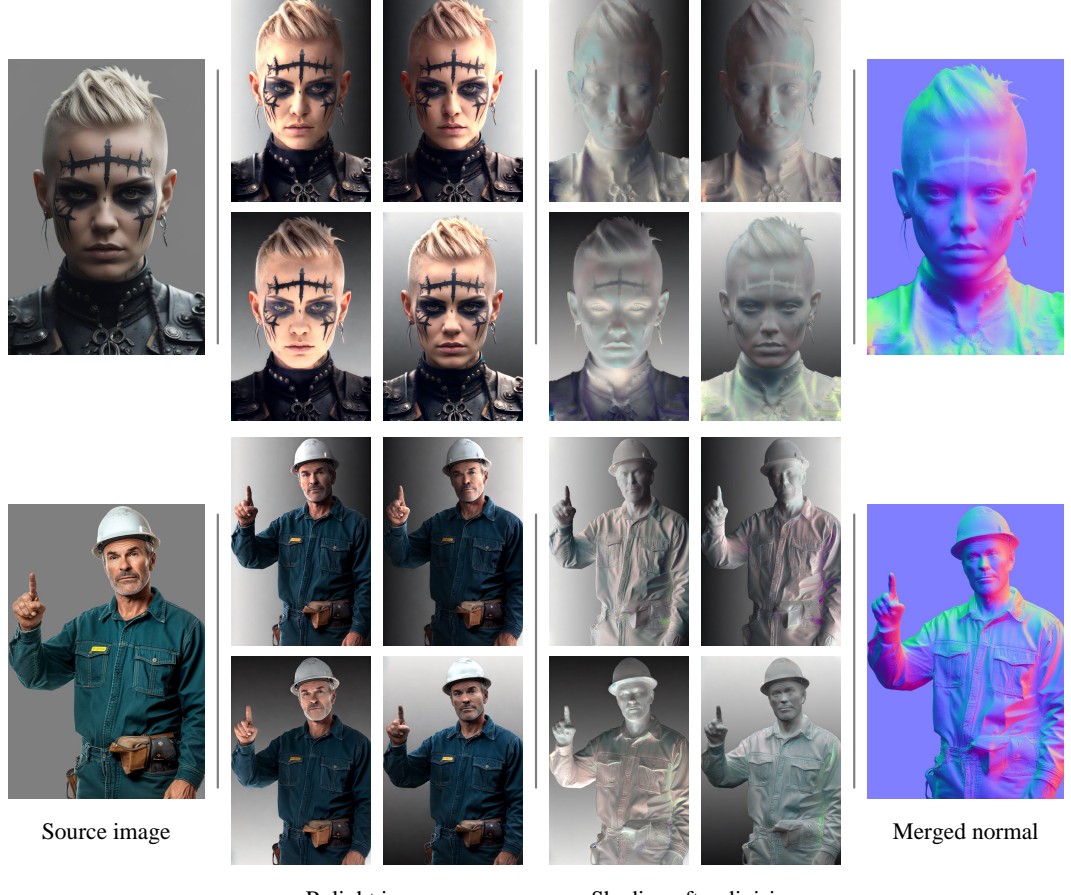

Source image      Relight images      Shading after division      Merged normal

Figure 12: **Blending relighting into normal maps.** The relight results from special illumination conditions (vertical or horizontal light sources) can be blended into normal maps. Note that the model is not trained on any normal estimation data. This indicates that the model can support a range of light stage applications that requires consistent illumination.