# OpenReview forum: "Scaling In-the-Wild Training for Diffusion-based Illumination Harmonization and Editing by Imposing Consistent Light Transport"
_ICLR.cc/2025/Conference — ICLR 2025 Oral_

### Official Review · Reviewer_djxF · 2024-10-20

**Soundness:** 3
**Presentation:** 4
**Contribution:** 4
**Rating:** 10
**Confidence:** 4

**Summary:**

The paper presents a diffusion-based method for image illumination control. Compared to vanilla T2I diffusion models, the proposed model additionally inputs a degraded or altered lighting image as a reference and an HDR environment map as lighting condition, and outputs the image under the target lighting. The key of the method is the incorporation of lighting consistency loss and the use of a variety of data, including 3d renderings, processed in-the-wild data, and OLAT light-stage data. The proposed method scales well on both large-scale data and different base models and advances other methods both qualitatively and quantitatively. The authors also provided additional applications, like background-conditioned illumination control and normal estimation.

**Strengths:**

The paper is in general very good, with many strengths:
1. The data preparation pipeline is very solid
    1. The authors assembled data from diverse sources for training, significantly enhancing the model's generalizability.
    2. For in-the-wild data, the authors utilized various intrinsic estimation methods, making the pipeline robust and avoiding inductive bias from any single method.
2. The proposed lighting consistency loss is novel and effectively prevents training degradation.
3. The training strategy demonstrates excellent scalability, is applicable to various diffusion base models and is capable of handling large-scale in-the-wild images.
4. The method achieves state-of-the-art image quality among diffusion-based lighting control methods.
5. The method also supports alternative lighting control interfaces, such as background condition.
6. The lighting consistency can be validated through the normal estimation application.

**Weaknesses:**

Some small weaknesses exist in the paper:

Formulation:
1. Only linear HDR images have light transport additivity, but natural images and images generated by diffusion models are usually tone-mapped LDR images, which do not have additivity. How is this misalignment handled?
2. There is material ambiguity in the single input image as different material-lighting combinations can yield a similar appearance. Hence I wonder how much the prompt and seed affect the model’s material estimation / final lighting result.

Implementation details:
1. Lighting consistency loss:
    1. To add the lighting consistency regularization during the training of latent diffusion models, the authors replaced the sum of two RGB-space images with an MLP that operates in latent space. The MLP works as a soft proxy, but does this MLP really work as intended without any other regularization?
    2. Also, the loss is derived using x_0 prediction, which is not very stable and can have low SNR when t is large, is there any special design regarding this issue?
2. Normal estimation:
    1. The formulas for normal estimation are not correct. Equation 8 should be minus rather than average. It is also not proper to call these two directions N_green and N_red. They should be defined in the world space with range [-1, 1] and visualized as (N + 1) / 2.
    2. It is better to use a consistent coordinate for the normal maps. In Fig (6), the man in the middle row has a different normal map specification from the examples on the last row, with R and G representing opposite directions in these two examples.

I would raise my score if the above problems were explained.

Besides, in the image results:
1. Some specular highlight seems to be baked in or show a strong correlation to the input image. E.g. the specular highlight on that car in Fig. 1, and on the helmet in Fig. 4
2. Some images have wrong global illumination (GI) effects, e.g. the ice-cream example on supp material page 6, has two shadows in opposite directions.

**Questions:**

I also have some small and less important questions:

Implementation details:
1. The reliability of DiffusionLight depends on the FOV of the image, is there any special handling on small FOV images?
2. To estimate environment maps from in-the-wild images, the original image is divided by albedo to obtain a shading map, and then projected to form the environment map: how to handle visibility/occlusion? e.g. on a concave object. Also, if the original image itself is already over-exposed, it will produce a wrong environment map even with HDR.
3. L242, the authors use CLIP for lighting filtering. However, as the text prompt is not enough for the lighting description, I wonder if the CLIP really has a good embedding for these lighting-related keywords. How effective does this filter work? Will something like an aesthetic score help here?
4. L246, do you mean “image-space” rendering? Does this image-based rendering pipeline consider GI? E.g. self-occlusion and inter-reflection. If not, is this the cause of some wrong GI effects?
5. For background conditioned inference, when only providing background, is env map L set to 0? Or is an environment map estimated from the background condition?
6. For normal estimation, when generating a shadow map, is 3 channels averaged? I can see some color in Fig 5 (c)’s shadow maps (which should be monocular), but it’s understandable for diffusion models. Besides, will specular highlights degrade this estimation? I'm also a little confused about the division by two in equation 8 and the way of ambient computation (using the average of four images instead of using the minimum value of all images). Can you clarify this?

Extra results:
1. Normal quantitative results: how well does the normal align with a real 3D normal map? Or what does the shape look like after running a normal integration on the estimated normal map?
2. How well is the temporal consistency when continuously changing the environment map L during inference?

Extra question:
1. Do you try to also impose lighting consistency on foreground and background?

I would also be happy to raise my scores if the authors provide answers or results regarding these questions.

---

> ### Author Response · Authors · 2024-11-21
> **Author Responses (1/n)**
>
> Thank you for the careful and constructive comments in this evaluation effort. We first discuss some considerations in the light transport consistency and then discuss each question in detail.
>
>
> ### MLP in light transport consistency (LDR vs. HDR, functionality as proxy, stability in x0 interpolation for extreme tSNR)
>
>
> Thanks for the question about LDR vs. HDR. This is indeed an important consideration. Actually in our work the challenge is a bit beyond LDR vs. HDR domains and extends to handling the images in latent space. The MLP acts as a bridging component, learning an implicit conversion that allows for smooth appearance interpolation in any of those domains (LDR / HDR / latent). As a result, the MLP is always necessary whenever the training data are not HDR images, no matter whether the diffusion model is pixel diffusion or latent diffusion. We will revise the related expositions.
>
>
> We also appreciate the insight to view this consistency as a proxy for numerical sum, and want to point out that the consistency is a bit more than "without any other regularization". The consistency is learned together with vanilla diffusion objective (Eq.6 in main paper) and the vanilla diffusion objective is a strong regularizer that makes sure that the model output follows the desired diffusion objective type. The consistency itself does not have any diffusion objective in it and cannot be used independently to train a diffusion model.
>
>
> Finally, thank you for bringing up the stability in x0 interpolation for extreme tSNR. This is a really inspirational way to consider the problem. Lets consider a simple interpolation of
> $$f(k, x_a, x_b)=k x_a + (1 - k) x_b$$
> where $x_a$ and $x_b$ are two different $x_o$ estimated using two independent model inferences with different conditions. The stability in extreme tSNR can be seen as a case where the matrices of $x_a$ and $x_b$ have very large data range (or standard deviation) because they have been premultiplied by very large ddim alphas_cumprod (or divided by very small k-diffusion sigmas). The question is whether approximating a model's output to $f(k, x_a, x_b)$ will influence numerical stability.
>
>
> Some related experimental evidence is the wide existence of CFG-distilled models. For any real number $k > 0$, we have
> $$f(k, x_a, x_b)=k x_a + (1 - k) x_b = x_b + k (x_a - x_b)$$
> if $x_b$ is estimated from empty condition, then $f(k, x_a, x_b)$ is the exact CFG formulation on conditional $x_a$ with guidance $k$. If we train a model to approximate this $f(k, x_a, x_b)$, we will obtain a CFG-distilled model. Note that in this case, this numerical range of $k > 0$ is even more aggressive than that in our paper (between 0 and 1). Considering the widely available of CFG-distilled models like Flux-1.dev and many LCM/Lighting models (many timestep-distilled models are also CFG-distilled), the terminal SNR does not seem to influence the model quality when approximating $f(k, x_a, x_b)$.
>
>
> In our work, the $f(k, x_a, x_b)$ is not receiving $x_a$ and $x_b$ from positive/negative conditions like CFG, but receives two different illumination environment maps as conditions. This should empirically produce an even smaller disparity between $x_a$ and $x_b$ and even less numerical instability than CFG distillation models. Considering that the tSNR instability is not frequently observed in CFG distillation literature, the interpolation for illumination conditions is likely to have similar stable training. We will add some related discussions to supplementary materials.

---

> ### Author Response · Authors · 2024-11-21
> **Author Responses (2/n)**
>
> (1) *Material ambiguity in the single input image*
>
>
> When the input images contain ambiguities (like materials), the final effect is jointly influenced by the base model, training data, and user prompts. Empirically, we find that stronger base model like Flux and larger, more diverse, higher quality training data are helpful for learning a more accurate mapping (and more responsive user edits from prompts). We will revise the related exposition.
>
>
> ---
>
>
> (2) *Revising normal estimation formulation and coordinate system*
>
>
> Thanks for pointing out this! The Eq 8 should be minus indeed. We are also revising the coordinate systems.
>
>
> ---
>
>
> (3) *Potential artifacts in the image results*
>
>
> Thanks for the observations and we are revising some related discussions.
>
>
> ---
>
>
> (Q1) *Environment map FOV handling and visibility/occlusion handling*
>
>
> In this work, all data on the input side are from methods chosen randomly (and/or with randomized degradations) as data augmentation to make the model more robust, whereas all data on the output side are of the highest quality as possible, like original raw in-the-wild images filtered by CLIP metrics. All randomness and errors on the input-side data are seen as data augmentation so that the model will work even when the input-side distribution is not strictly aligned with the distribution of real-world inputs.
>
>
> Regarding illumination environment maps's FOV handling, though methods like DiffusionLight does include "generative" patterns for small FOV images, we place a stronger emphasis on the correspondence between the environment lighting map and the pixel colors in the original image. For example, we also use an environment-from-normal approach to project shading colors with their normal direction to a environment ball (supp, L68-74). Using mixed environment map estimators helps the cases when FOV correspondences are relatively weak.
>
>
> The visibility and occlusion relationships between objects are handled with similar principles to allow certain amounts of error and noise to serve as data augmentations for training robust models. For example, in the environment-from-normal approach, if a position in the normal sphere is mapped with multiple colors from different pixels in the shading maps, we take averages. While this may introduce some physical inaccuracies (it can be less accurate than explicitly modeling occlusion/visibility), it is effective in establishing a learnable correspondence between the lighting map and pixel values, and the extra randomness may also facilitate model robustness. We will add related discussions.
>
>
> ---
>
>
> (Q3) *CLIP for sample filtering*
>
>
> The CLIP-filtering is mainly for getting rid of data that has no relationship to illumination. For example, most in-the-wild text-image datasets have large numbers of flat vector graphs, icons, UI designs, logos, and so on. Performing intrinsic decomposition on those data are not meaningful for our problem and those data need to be removed. Although CLIP vision is not perfect in text-image alignment, in our tests, it is already powerful enough to get rid of almost all those non-related data.
>
>
> ---
>
>
> (Q4) *Image-based rendering pipeline*
>
>
> The image-based rendering pipeline prioritizes speed (using pure pytorch matrix dot products) to generate more diverse illumination appearances over rendering quality. This is mainly because the rendered images are only used as degradation images, and their distribution diversity and quantity is much more important than individual quality and fidelity. The image-based rendering pipeline does not include global illumination or inter-reflection modeling between objects. However, it does have random hard shadows (L239).

---

> ### Author Response · Authors · 2024-11-21
> **Author Responses (3/n)**
>
> (Q5) *Background conditioned inference*
>
>
> When using the background as a condition, the environment branch can be zeroed out (by completely removing the connection to the neural network and not only setting environment as zero), similar to [1]. We plan to release the models with and without the environment branch for various use cases.
>
>
> [1] Relightful Harmonization: Lighting-aware Portrait Background Replacement
>
>
> ---
>
>
> (Q6) *Averaged channels for normal estimation and influence of specular terms*
>
>
> For generating normal maps, we do average all color channels to calculate the intensity. Handling specular terms may involve unique considerations. For example, when comparing shading maps (without hard shadow) under left-to-right versus right-to-left lighting, the two maps should ideally be exact negatives of each other in log space. Any discrepancies or offsets between the two maps often indicate specular reflections. Such offsets could potentially be extracted as specular components using more sophisticated formulations. We are also working on exploring more advanced methods to improve the decomposition formulations.
>
>
> ---
>
>
> (Q7) *How well does the normal align with a real 3D normal map?*
>
>
> In our study, the normal maps computed by dividing appearances visually resemble more to those captured via light-stage photography metrics (like those normals in [2]) than those from 3D-rendered normals (like Objaverse normals). This is likely due to the hard mesh boundaries (though high-quality 3d normals would have internal local normal textures) and less detailed hair/fur in 3D normals. We find that the appearance of the division-based normals aligns more closely with empirical measurements from light stage setups.
>
>
> [2] Total Relighting: Learning to Relight Portraits for Background Replacement
>
>
> ---
>
>
> (Q8) *Temporal consistency in changing the environment map*
>
>
> Temporal consistency is not addressed in this work, and as a result, the model may not produce explicitly optimized temporal coherence when changing conditions. We are taking a close look at latest video models like CogVideo [3] and actively investigating the potential to train IC-Light on those models. For example, recently dimension-X [4] showed that video models can be tuned to produce a short video representing scene rotation, this is likely to indicate that illumination transitions like environment map rotations can be learned in a similar way. This is one promising way to consider future experiments.
>
>
> [3] CogVideoX: Text-to-Video Diffusion Models with An Expert Transformer
>
>
> [4] DimensionX: Create Any 3D and 4D Scenes from a Single Image with Controllable Video Diffusion
>
>
> ---
>
>
> (Q9) *Imposing lighting consistency on foreground*
>
>
> Light transport consistency is masked by foreground mask $M$ in Eq.4 and Eq.5.

---

> > ### Comment · Reviewer_djxF · 2024-11-22
> >
> > Thanks for the explanation, my questions regarding the lighting consistency loss and low SNR problem are solved. I also appreciate the authors' clarification on material ambiguity and other questions (1-5, 8). I will raise my score as I have promised.
> >
> > I still have some questions regarding normal (Q6, Q7) and foreground-background lighting consistency (Q9).
> >
> > Regarding Normal Estimation:
> > Q6: It can be better to include specular cases as a limitation in the normal estimation part
> > Q7: The normal estimator in the total relighting paper is actually trained with GT/acquisited normal obtained through the photometric stereo with a "per-pixel over-determined system". If I understand correctly, the normal estimation here is more of an empirical method instead of a fully determined/physical-based one. Hence, I'm interested to see how well the empirical normal estimation can be without explicit normal supervision like total relighting, as well as a way to measure the lighting consistency quantitatively. I guess existing datasets in the photometric stereo field would be a good way to test if synthetic 3D data is not suitable.
> >
> > Regarding foreground & background consistency:
> > Q9: Sorry for not being clear in my question, I was hoping to see some lighting consistency involving light transport on both foreground and background. E.g. the foreground object can cast shadows on the background scene.

---

> > > ### Author Response · Authors · 2024-11-23
> > > **Author Responses**
> > >
> > > Thanks for the suggestions! Indeed, the normal extraction is an empirical method since the neural models are not trained to approximate determined ground truths like light stage photography metrics or 3D normal maps. This should be noted in the manuscripts, and we have added related expositions. Thanks for this!
> > >
> > > About regulating backgrounds during training – in this work we are not applying specific regulations to backgrounds, and the models’ capability for background is similar to diffusion-based inpaint model (like generating a background with shadow behind objects when necessary). On the other hand, scene-level illumination editing is a really challenging open problem and deserves more future research attention, with many special considerations like object interreflection, hard shadows, light source inside scene, etc.

---

### Official Review · Reviewer_Nsxu · 2024-11-02

**Soundness:** 4
**Presentation:** 3
**Contribution:** 4
**Rating:** 10
**Confidence:** 3

**Summary:**

This paper proposes a method to generate illumination and background from given foreground reference images and text prompts. The paper's key contribution is to use a new consistency loss, which makes the foreground appearance consistent under different lighting conditions. The paper argues that thanks to the new loss functions, the fine-tuning of diffusion models can be scaled up to over 10 million samples. Experimental results show both qualitatively and quantitatively superior image editing results compared to existing methods.

**Strengths:**

### Loss design
The design of the new loss function is reasonable. Adding the appearance consistency under a mixed lighting environment is somewhat inspired by the mixing strategies in augmentation and regularization in neural network training, while using that regularization for image illumination editing would be new to me.

### Large-scale training
The paper fine-tunes major image generation models with over 10 million samples, resulting in an immediately useful tool for image editing.

**Weaknesses:**

### Unclearness of the releases of the data and implementation
From the paper, it is unclear whether the authors will release the data and implementations or intend to provide a black box web tool for users.

**Questions:**

I consider that the proposed method and the resulting tool will be immidiately useful for many users. I would like to ask the authors how it will be released.

---

> ### Author Response · Authors · 2024-11-21
> **Author Responses**
>
> Thank you for the insights in the evaluation and the appreciation in the application of image illumination editing tool.
>
> With regard to the model and data, our current plan of the release timeline is (1) SD1.5 models and demos, and then (2) demos for latest models like Flux, and then (3) more ablative models for SD1.5 and some related data, and then (4) more weights and inference codes for latest models like Flux, and then (5) more ablative models for latest models like Flux, and then potentially (6) more related training implementations and data processing. We are also considering having a standard standalone software.

---

### Official Review · Reviewer_2MNV · 2024-11-02

**Soundness:** 4
**Presentation:** 3
**Contribution:** 4
**Rating:** 10
**Confidence:** 5

**Summary:**

This paper proposes a physically inspired relighting scheme for training a lighting-aware generative model. The idea of imposing consistent light (IC-Light) is based on the observation of the real world; i.e., an object’s appearance under different illumination conditions is consistent with its appearance under mixed illumination. To this end, the authors train a generative model conditioned by lighting environment maps and an additional light consistency loss. In order to train the model effectively, the authors introduce several strategies for data collection, including rendering intrinsic components from a synthetic dataset, augmenting the existing dataset with random shadows, and separating the foreground and background. They manage to collect more than 10 million images for final training. The results of the proposed method appear to be astonishing. The authors also demonstrate side applications such as image harmonization and surface normal estimation, both of which are object-centered.

**Strengths:**

- The results are very strong. The performance of the proposed IC-Light is astonishing, and it can be further improved with a stronger generative model as the backbone.

- The method is sound. Although the fundamental observation (the intrinsic component is always invariant to lighting) is not new, the authors are able to effectively leverage this observation in the training of generative models. They thoroughly explain all the details and intuition behind their method.

- The contribution is significant, impacting not only the vision and graphics communities but also holding great commercial potential for the photography and movie industries.

**Weaknesses:**

- The fundamental observation of this paper is common knowledge in the computational photography and low-level vision communities. Unfortunately, I found that the authors overlooked a large portion of the literature in these fields, especially in terms of intrinsic image decomposition. I noticed that the authors use several IID methods to generate the intrinsic components and mention them in the supplemental materials; I think the authors should also list them in the main paper.

- The quantitative experiments are somewhat limited (including details on the rendering data, e.g., how distinct they are from those in training dataset). Additionally, the numerical and visual comparisons could be improved by including methods like Neural Gaffer.

- Some figures in the paper should be better explained:  For Figure 4 and 5-(c), what is the prompt/condition for the background/lighting?

- L. 427 and L. 431: " shadow maps" should be shading maps.

**Questions:**

- Although the IC-Light scheme works very well for foreground relighting, would it perform equally well for general scene-level relighting? For instance, could the proposed method achieve effects similar to turning on or off a visible light source within a scene, as shown in the paper "Physically-Based Editing of Indoor Scene Lighting from a Single Image" by Li et al., ECCV 2022?

- In Figure 1, the authors demonstrate impressive lighting effects. I am curious whether the method consistently provides such high-quality, precise lighting effects, or if it requires careful tuning of the prompt and seed to achieve this result.

---

> ### Author Response · Authors · 2024-11-21
> **Author Responses**
>
> Thank you for the insights in the evaluation and the hints for revising manuscripts.
>
>
> ---
>
>
> (1) *More discussions for literature in computational photography and low-level vision communities, especially in terms of intrinsic image decomposition.*
>
>
> Thank you for this suggestion. We will include more discussions (a new section) about works in intrinsic images, computational photography, and related low-level vision methods.
>
>
> ---
>
>
> (2) *More quantitative experiments.*
>
>
> Thank you for this feedback. We are working on improving quantitative presentations, and will add dataset examples in the supplementary materials.
>
>
> ---
>
>
> (3) *Revising expositions for some figures and fixing some typos.*
>
>
> We will revise the manuscripts, captions, and figures according to the comments.
>
>
> ---
>
>
> (Q1) *What about general scene-level relighting?*
>
>
> Thank you for the insights about scene-level relighting. The scene-level illumination poses additional challenges, especially when accounting for internal light sources (that cannot be represented by environment maps), occlusions, and complex hard shadow interactions. These elements require more sophisticated data synthesis or inference pipeline. A possible future direction is decomposing the scene into objects/components, applying individual illumination effects, and blend components with shadow generation, inter-reflections, and other types of harmonization. This is definitely a promising and challenging open direction, and we will include it in the revision as well as references to the mentioned related paper.
>
>
> ---
>
>
> (Q2) *Consistency in high-quality lighting effects and hyper-parameters*
>
>
> The framework is designed to be robust to diverse user inputs and lighting conditions. Figure 1 is designed to display typical use cases like hard shadow (the woman), changing light direction (the man), transparency (the perfume), special effects (the neon car), etc. Consistent high-quality results are relatively easy to obtain in simple cases like altering the illumination color or direction (like the perfume bottles and man). For more complicated cases like obtaining a specific shape of hard shadow casted by Venetian blind patterns, more prompts are usually needed for achieving the very customized effects.

---

> > ### Comment · Reviewer_2MNV · 2024-11-26
> >
> > Thanks for the clarification and discussion. I appreciate the authors’ efforts in adding a new section about the intrinsic images to the main paper and thoroughly explaining the dataset creation in the supplemental. I believe all my concerns have been addressed. Therefore, I will raise my score.

---

### Official Review · Reviewer_Tpnw · 2024-11-03

**Soundness:** 4
**Presentation:** 4
**Contribution:** 4
**Rating:** 10
**Confidence:** 5

**Summary:**

This paper introduces an extremely well-generalized illumination control model for recent diffusion models. Illumination control is essential for numerous artistic effects and is key to driving the mode of generated imagery. Recent methods struggle to capture this control due to the lack of large-scale real-world datasets. The key idea of the paper is to use the linearity of lighting as a constraint, i.e., the blending of two images of the same scene under two different illuminations is the same as blending the two illuminations and applying them to the scene. The authors collect a high-quality dataset from various sources using numerous recent pre-trained models to generate a large-scale paired dataset. The effectiveness of the idea is extremely well-demonstrated both qualitatively and quantitatively. The resulting images have extremely consistent materials, even so consistent that high-quality normal maps can be extracted using multi-light generation.

**Strengths:**

This is an extremely well-written and well-motivated paper; I believe this is a really good example of a highlight paper. Key points I have considered during reviewing the paper:
* The paper proposes a well-working solution to an extremely challenging task. Lighting and material are only observed in a highly intertwined manner, making it difficult to control only one without changing the other. Previous methods tried to solve this problem with paired synthetic datasets, albeit falling into a domain gap.
* The proposed dataset provides a really important insight, which can be important for the community: using pre-trained models and synthetic degradations can still provide helpful signals.
 * The light transport consistency is a simple but very effective way to enforce consistency. However, the idea is not new, earlier methods required multi-light datasets to utilize a similar constraint. This paper shows a way how to realize it with in-the-wild samples.
* The approach is extremely well-documented together with practical insights and considerations.

**Weaknesses:**

This is an extremely good paper overall; I just have minor comments, which might help improve the writing quality.
* It would be great to highlight more which datasets are used for which parts of the training. Potential cause for confusion: the pipeline figure shows that the training required I_d, although I_d is used only for in-the-wild samples, but then it is not clear where the other components of the dataset are used.
* Some additional related works, which could be discussed:
  * [Shape-form-shading](https://www2.eecs.berkeley.edu/Pubs/TechRpts/2013/EECS-2013-117.pdf) - how it relates to the current normal extraction
  *  [LightIt](https://peter-kocsis.github.io/LightIt/) - illumination control with a generated paired dataset, where similarly an automatically relit image is used as input during training to avoid domain gap.
* It would be good to highlight the best metrics with bold in Tab. 1.

**Questions:**

It would be helpful for easier understanding to better describe the followings:
* More details would be helpful on the synthesis of L1 and L2 (L.319), probably even some visuals in the supplementary.
* More qualitative samples from the dataset with all the methods used to generate the synthetic components.

---

> ### Author Response · Authors · 2024-11-21
> **Author Responses**
>
> Thank you for the insightful comments and evaluation.
>
>
> ---
>
>
> (1) *It would be great to highlight more which datasets are used for which parts of the training.*
>
>
> For in-the-wild images, degradations are used as training inputs $I_d$. For 3D data and light stage data, degradations are not needed and we directly use random illumination environment maps to render random appearances as training inputs $I_d$ (L249). To be specific, 3d data are rendered by the rendering pipeline and light stage data are rendered by weighting one-light-at-time images. We will revise the paper to clarify this.
>
>
> ---
>
>
> (2) *Additional citations and discussions and better bold highlight in tables.*
>
>
> We will incorporate the additional references and clarify the tables with bold highlights in the revision.
>
> ---
>
>
> (Q1) *More details would be helpful on the synthesis of L1 and L2 (L.319), probably even some visuals in the supplementary.*
>
>
> Thank you for this suggestion and we are working on adding visualizations to the supplementary material.
>
>
> ---
>
>
> (Q2) *More qualitative samples from the dataset with all the methods used to generate the synthetic components.*
>
>
> We appreciate this suggestion and we are working on adding some dataset examples in the supplementary material.

---

> > ### Comment · Reviewer_Tpnw · 2024-11-22
> > **Thank you**
> >
> > Thanks a lot for the authors for their answers! I am looking forward to see the final revision!

---

### Public Comment · ~Jiahui_Li8 · 2024-11-19
**Questions about implementation details**

1.  Estimation environment maps： L-45 in supplementary material refers to "A environment-from-normal method. Probability is 70%. See the below section." However, the following text does not explain in detail.
2.  L-68: "We divide the original image by the albedo to obtain the a shading map." In the process of obtaining shadingmap, is it necessary to perform gamma reverse correction on the data？
3.  Unclear processing : L-57 to L-83 in supplementary material，the composition of new shading map including hard shadow, soft shadow and specular reflection is discussed, and finally, the new shading map is multiplied with albedo to obtain the "degradation" is acquired. However, the specific synthesis process and theoretical model are not explained in detail. Like "projected onto the normal sphere through pixel normal direction" at L-69. Please be detailed.

---

> ### Author Response · Authors · 2024-11-21
> **Thanks for discussion**
>
> Thanks for the public comment. The “below section” around supplement L44 refers to the section around supplement L67-73. The shading map computation does not use gamma correction, but image rendering often have a step from HDR images to LDR and that step may have an effect similar to gamma correction. About multiplying intrinsic components and processing normal vectors, a good tutorial is “An L1 Image Transform for Edge-Preserving Smoothing and Scene-Level Intrinsic Decomposition” from Bi et al, with many fundamental tutorials. Another great tutorial is “Shape, Albedo, and Illumination from a Single Image of an Unknown Object” from Barron et al. 2012 for processing environments with image-space shading and normals

---

> > ### Public Comment · ~Jiahui_Li8 · 2024-11-22
> > **question about the environment-from-normal method**
> >
> > Thank you very much for your reply and study guidance!
> > I still have a question about the environment-from-normal method: Is the obtained environment-from-normal reasonable according to the section around supplement L67-73? I wonder if the new shading map rebuilt with the resulting environment map is consistent? If not, can the environment map obtained by this method really be used as an "environment"?

---

### Author Response · Authors · 2024-11-23
**File Upload**

We thank all reviewers for the constructive and insightful efforts in evaluating this work. We have uploaded revised files, with several modifications:

1. a new section in related work for intrinsic images;
2. more visualizations and examples in the supplement materials, including shadow examples, degradation, environment map decomposition;
3. revised expositions in the methodology for MLP in light transport consistency and normal computation.

The above points are marked in dark blue (in both main paper and supplementary materials).

Besides these points, we have also revised typos, references, figure captions, formulations, and all other minor points mentioned in the reviewing process.

Thanks again for the constructive efforts in the comments and reviews.

Authors

---

> ### Public Comment · ~Christian_Richardt1 · 2024-11-24
> **Paper feedback**
>
> Thank you for the interesting paper and the cool results!
>
> Here are some things I noticed that you might want to fix in the final version of the paper. I hope this helps:
> * lines 289 and 290: the number 2307 should probably be $2304 = 3 \cdot 768$
> * line 296 (Eq. 1): $\mathcal{E} \to \boldsymbol\varepsilon$; $t$ should not be bold as it's a scalar; delete one extra closing parenthesis
> * line 335 (Eq. 4): missing closing parenthesis inside the norm
> * line 337: delete one “same”
> * line 341: (Eq. 5): $\mathcal{E} \to \boldsymbol\varepsilon$; $t$ should not be bold as it's a scalar; delete one extra closing parenthesis in the middle; the 1/2 subscripts for $\mathbf{L}$ should not be bold
> * line 347: $\lambda_\text{relight} \to \lambda_\text{vanilla}$
>
> Please consider discussing [IntrinsicDiffusion (SIGGRAPH 2024)](https://intrinsicdiffusion.github.io/) in your new related work section on intrinsic images; it seems quite relevant.
>
> It would also be good to clearly define the projection format of the lighting environment (which direction each pixel maps to) as square aspect ratios are unusual.

---

> > ### Author Response · Authors · 2024-11-26
> > **Thanks for the suggestions**
> >
> > Thanks a lot for the suggestions and public comments! We have addressed mentioned points, references, and added more descriptions to supp Fig.4 about environment map.

---

### Public Comment · ~Wang_Zhen1 · 2024-11-25
**Questions about the details of data construction**

Estimation environment maps: Figure 3 (L118) in supplementary material shows some images about environment map(L), but how to get these images?

I think the pipeline to get these environment map(L) can be described below: firstly, take the output from DiffusionLight(means 1 or 2), then do the filter to blur these images, finally resize filtered images to the designated size.

 The supplementary material does not explain how to get environment map(L) in detail.Please be detailed about how to get environment map(L) in  Figure 3 (L118).

---

> ### Author Response · Authors · 2024-12-03
> **Thanks for the public comments!**
>
> Thanks for the comment! In the latest version, some expositions should relate to this: (see also new supp.fig.4) ... A typical full environment map is usually of ratio 2:1, with size 64x32 when convoluted. We use the front half (facing the image) of the convoluted environment map, which is 32x32. Using the front half makes normal-based environment extraction easier (since the image-space normals often do not have any pixels facing to the back half) ...

---

### Public Comment · ~Li_Niu2 · 2024-11-29
**Related Works on Image Harmonization and Image Composition**

This is a great work! It is suggested to add more discussions on image harmonization https://github.com/bcmi/Awesome-Image-Harmonization and image composition https://github.com/bcmi/Awesome-Image-Composition in the related works.

---

> ### Author Response · Authors · 2024-12-03
> **Thanks for the public comments!**
>
> Thanks for providing the additional sources for references!

---

### Public Comment · ~Jiahui_Li8 · 2024-12-02
**Questions about image_based_lighting:**

Questions about image_based_lighting:
1. In the supplementary material, the description of creating shadow_map with normal is not clear. Are the terms' shadow map 'and' shading map 'problematic in this paragraph?
2. in the preparation of In-the-Wild dataset, albedo_map, normal_map, env_map[HDR], shadow_mask, shadow_map were obtained, but how was the subsequent IBL implemented? Which reflection model is used?

I would appreciate it if you could provide details or know me!

---

> ### Author Response · Authors · 2024-12-03
> **Thanks for the public comments!**
>
> Normal map and environment map will create a shading map (that can be applied to appearances) with that environment, and yes, it is better to always call it shading map. Hard shadow are from shadow materials, whereas soft shadows are from normal and environment maps. About randomized specularity, the synthesizing is by increasing the specular of random areas (for Phong it is Specular term; for Cook-Torrance and other bsdf one can also use roughness and/or metallic etc).

---

### Public Comment · ~Sergio_Sancho1 · 2024-12-02
**Images in the loss of Eq. 5**

Congratulations on the amazing work! I have a question regarding the loss in Eq. 5. If I understand correctly, $\boldsymbol{L}_1$ and $\boldsymbol{L}_2$ are obtained by randomly masking $\boldsymbol{L}$. However, the equation seems to also involve $\boldsymbol{\varepsilon}(\boldsymbol{I}\_{\boldsymbol{L}\_{1}})\_{t}$ and $\boldsymbol{\varepsilon}(\boldsymbol{I}\_{\boldsymbol{L}\_{2}})\_{t}$. I am curious about how to get the images $\boldsymbol{I}\_{\boldsymbol{L}\_{1}}$ and $\boldsymbol{I}\_{\boldsymbol{L}\_{2}}$, since $\boldsymbol{L}_1$ and $\boldsymbol{L}_2$ are generated on the fly. Am I missing something? Thank you!

---

> ### Author Response · Authors · 2024-12-03
> **Thanks for the public comments!**
>
> Thanks for this comment! This is a typo and should be fixed soon: The $I_{L_1}$ and $I_{L_2}$ in Eq. 5 should be $I_{L}$ (The “1” and “2” are typo). And here $L=L_1+L_2$. (see also Fig.3-(b)) Thanks again for finding this!

---

> ### Public Comment · ~Tianyi_Zhang27 · 2024-12-03
>
> Congratulations on the amazing work! I have a confusion about derivation of line 324 in page 7. If the $I_{L_1}$ and $I_{L_2}$ in Eq. 5 is $I_{L}$, we apply Eq. (3) as $I_{L_1+L_2} = I_{L_1} + I_{L_2}$ and get the following equations:
> \begin{align}
> \frac{I_{\sigma_t}^{L} - \epsilon_{L_1+L_2}}{\sigma_t} &= \frac{I_{\sigma_t}^{L} - \epsilon_{L_1}}{\sigma_t} + \frac{I_{\sigma_t}^{L} - \epsilon_{L_2}}{\sigma_t} \\\\
> I_{\sigma_t}^{L} - \epsilon_{L_1+L_2} &= I_{\sigma_t}^{L} - \epsilon_{L_1} + I_{\sigma_t}^{L} - \epsilon_{L_2} \\\\
> \epsilon_{L_1+L_2} &= \epsilon_{L_1}  + \epsilon_{L_2} - I_{\sigma_t}^{L}
> \end{align}
> which conflict to the $\epsilon_{L_1+L_2} = \epsilon_{L_1}  + \epsilon_{L_2} $ in line 325. Could you please explain my little confusion? Thank you!

---

> > ### Author Response · Authors · 2024-12-03
> > **Thanks for discussion**
> >
> > Thanks for the comments. We also noticed that some diffusion objective (like eps-prediction) will introduce an extra scaling factor $k$ so that $k I_{L_1+L_2}=I_{L_1}+I_{L_2}$ and the $k$ is the exact number of blended elements (in this case $k=2$). But other diffusion objectives like $x_0$-prediction are not influenced. Similar to the HDR/LDR scaling, this scaling is also implicitly learned by MLP. This discussion should also be added to the appendix/supplement.
> >
> > For example, when blending $k=2$ objectives
> >
> > $k I_{L_1+L_2}=I_{L_1}+I_{L_2}$
> >
> > $k\frac{I_{\sigma_t}^L-\epsilon_{L_1+L_2}}{\sigma_t} =\frac{I_{\sigma_t}^L-\epsilon_{L_1}}{\sigma_t}+\frac{I_{\sigma_t}^L-\epsilon_{L_2}}{\sigma_t}$
> >
> > $kI_{\sigma_t}^L-k\epsilon_{L_1+L_2} =I_{\sigma_t}^L-\epsilon_{L_1}+I_{\sigma_t}^L-\epsilon_{L_2}$
> >
> > $k\epsilon_{L_1+L_2} = \epsilon_{L_1}+\epsilon_{L_2}$
> >
> > So that the MLP would implicitly scale the data by $1/k$. Note that using $\{I^{L}\_{\sigma_t}, I^{L_1}\_{\sigma_t}, I^{L_2}\_{\sigma_t}\}$ or not makes no difference. Even in ideal cases where $\{I^{L}\_{\sigma_t}, I^{L_1}\_{\sigma_t}, I^{L_2}\_{\sigma_t}\}$ are all available and applied, with ground truth noise $\epsilon$, and $k=2$, we still have
> >
> > $k I_{L_1+L_2}=I_{L_1}+I_{L_2}$
> >
> > $k\frac{I_{\sigma_t}^{L_1+L_2}-\epsilon_{L_1+L_2}}{\sigma_t} =\frac{I_{\sigma_t}^{L_1}-\epsilon_{L_1}}{\sigma_t}+\frac{I_{\sigma_t}^{L_2}-\epsilon_{L_2}}{\sigma_t}$
> >
> > $kI_{\sigma_t}^{L_1+L_2}-k\epsilon_{L_1+L_2} =I_{\sigma_t}^{L_1}-\epsilon_{L_1}+I_{\sigma_t}^{L_2}-\epsilon_{L_2}$
> >
> > $k(I_{L_1+L_2} + \sigma_t \epsilon) -k\epsilon_{L_1+L_2} =(I_{L_1} + \sigma_t \epsilon)+(I_{L_2} + \sigma_t \epsilon)-(\epsilon_{L_1}+\epsilon_{L_2})$
> >
> > $I_{L_1}+I_{L_2} + 2\sigma_t \epsilon -k\epsilon_{L_1+L_2} =I_{L_1} + \sigma_t \epsilon+I_{L_2} + \sigma_t \epsilon-(\epsilon_{L_1}+\epsilon_{L_2})$
> >
> > $k\epsilon_{L_1+L_2} = \epsilon_{L_1}+\epsilon_{L_2}$
> >
> > So that using $I^{L}\_{\sigma_t}$ or $\{I^{L_1}\_{\sigma_t}, I^{L_2}\_{\sigma_t}\}$ makes no difference to the result. And the MLP would implicitly learn to scale the data by $1/k$.
> >
> > And this only affect some objectives like eps, while other objectives like $x_0$ does not need this conversion.

---

> > > ### Public Comment · ~Tianyi_Zhang27 · 2024-12-03
> > >
> > > Thank you for your reply.
> > >
> > > I still have the following two questions and hope you can provide further guidance.
> > > 1. Although introducing an extra scaling factor $k$ can make $k I_{L_1+L_2} = I_{L_1} + I_{L_2}$, it also introduces a conflict of the Eq. (3), $I_{L_1+L_2} = I_{L_1} + I_{L_2}$, which is validated by real-world measurements.
> > >
> > > 2. As far as I know, the $x_0$ remains constant during diffusion, which means the input $x_t$ of the denosing network should satisfy $x_t=I_{\sigma_t}^{L_i} = \sigma_t I_{L_i} + \epsilon$. It indicates that the $I_{L_1}$ and $I_{L_2}$ in Eq. (5) can not be replaced by $I_{L}$ with any diffusion objective.
> > >
> > > Perhaps these confusions stem from my limited knowledge, and I am looking forward to your further guidance.

---

> ### Author Response · Authors · 2024-12-03
> **Thanks for discussion**
>
> Thanks for the comments. The $I_{L_1+L_2}=I_{L_1}+I_{L_2}$ measured by real-world validation is for HDR images. The images for training are LDR images and need MLP as an implicit conversion to make use of this equivalence. Scaling HDR images by any global scaler does not influence its LDR appearance (if tone-mapping has intensity pre-norm).
>
> When denoising with Fig.3 (b), the clean latents for the two $L_{1,2}$ are unknown. Fig.3 (b) should be seen as a big model as a whole to denoise noisy latents from known clean latents $I_L$ using conditions $I_d, L_1, L_2$ and prompts.

---

### Meta-Review · Area_Chair_6Shg · 2024-12-21

**Metareview:**

This paper presents an illumination control model for relighting foreground objects using image diffusion models. The proposed model allows users to adjust lighting conditions flexibly using either text descriptions or background images. The paper introduces effective strategies for collecting paired illumination datasets to support the training process. A notable contribution is the physically-grounded light transport consistency loss, which enforces consistent intrinsic properties across lighting. The model achieves impressive results, preserving intrinsic object properties under diverse lighting conditions. The paper addresses a challenging and impactful problem by offering a novel, physics-based solution with impressive results.

**Additional Comments On Reviewer Discussion:**

The paper received highly positive feedback in the reviews, with some concerns addressed during the rebuttal process. The paper focuses on relighting and relates to computational photography and low-level vision, particularly intrinsic image decomposition. However, these topics were initially missing from the literature survey. In response, the revision has incorporated more extensive discussions of related work.

Reviewers expressed curiosity about the model's applicability to scene-level relighting. The rebuttal clarified that scene-level relighting poses more significant challenges and outlined potential future research directions. The proposed method relies on the linearity of light transport, but a reviewer raised a concern regarding the impact of nonlinear adjustments commonly applied to images. The rebuttal addressed this by explaining that the proposed MLP effectively handles such issues within the latent domain. Additionally, questions about the formulation, lighting consistency loss, and normal estimation were raised, all of which were satisfactorily addressed in the rebuttal.

---

### Decision · Program_Chairs · 2025-01-22

Accept (Oral)